# Identification of abscission checkpoint bodies as structures that regulate ESCRT factors to control abscission timing

Lauren K Strohacker[1,2], Douglas R Mackay[1], Madeline A Whitney[1], Genevieve C Couldwell[1], Wesley I Sundquist[2]*, Katharine S Ullman[1]*

[1]Department of Oncological Sciences, Huntsman Cancer Institute, University of Utah, Salt Lake City, United States; [2]Department of Biochemistry, University of Utah School of Medicine, Salt Lake City, United States

**Abstract** The abscission checkpoint regulates the ESCRT membrane fission machinery and thereby delays cytokinetic abscission to protect genomic integrity in response to residual mitotic errors. The checkpoint is maintained by Aurora B kinase, which phosphorylates multiple targets, including CHMP4C, a regulatory ESCRT-III subunit necessary for this checkpoint. We now report the discovery that cytoplasmic abscission checkpoint bodies (ACBs) containing phospho-Aurora B and tri-phospho-CHMP4C develop during an active checkpoint. ACBs are derived from mitotic interchromatin granules, transient mitotic structures whose components are housed in splicing-related nuclear speckles during interphase. ACB formation requires CHMP4C, and the ESCRT factor ALIX also contributes. ACB formation is conserved across cell types and under multiple circumstances that activate the checkpoint. Finally, ACBs retain a population of ALIX, and their presence correlates with delayed abscission and delayed recruitment of ALIX to the midbody where it would normally promote abscission. Thus, a cytoplasmic mechanism helps regulate midbody machinery to delay abscission.

*For correspondence:
wes@biochem.utah.edu (WIS);
katharine.ullman@hci.utah.edu
(KSU)

**Competing interest:** See
page 20

## Introduction

Checkpoints function throughout the cell cycle to ensure accurate and timely coordination of cell cycle events (*Hartwell and Weinert, 1989*). The abscission (NoCut) checkpoint is active during cytokinesis when cells are connected by a narrow intercellular bridge containing a microtubule-rich structure termed the midbody. Abscission checkpoint signaling ceases if no errors are detected, reflecting checkpoint satisfaction. However, if residual mitotic errors are detected, then the checkpoint remains unsatisfied and the final cut of cytokinesis is delayed (*Norden et al., 2006*; *Steigemann et al., 2009*). To date, this checkpoint has been shown to be responsive to four error conditions: chromatin bridges within the midbody, reduced levels of particular nuclear pore proteins, tension at the intercellular bridge, and previous DNA replication stress (*Lafaurie-Janvore et al., 2013*; *Mackay et al., 2010*; *Mackay and Ullman, 2015*; *Petsalaki and Zachos, 2019*; *Steigemann et al., 2009*). Left unchecked, these conditions have the potential to disrupt new daughter cell functions. This is most clear in the case of chromatin bridges, which can break under tension forces in the absence of the abscission checkpoint (*Petsalaki and Zachos, 2021*) and even break and re-fuse in the absence of p53, ultimately leading to chromothripsis, a mutational signature commonly found in cancer genomes (*Maciejowski et al., 2015*; *Umbreit et al., 2020*). Loss of the abscission checkpoint accelerates the time to abscission and induces genomic instability in cultured cells (*Sadler et al., 2018*). Thus, the abscission checkpoint appears to protect cells from accumulating damage arising from mitotic errors, which otherwise have the potential to promote tumorigenesis.

**eLife digest** When a cell divides, it must first carefully duplicate its genetic information and package these copies into compartments housed in the two new cells. Errors in this process lead to genetic mistakes that trigger cancer or other harmful biological events.

Quality control checks exist to catch errors before it is too late. This includes a final 'abscission' checkpoint right before the end of division, when the two new cells are still connected by a thin membrane bridge. If cells fail to pass this 'no cut' checkpoint, they delay severing their connection until the mistake is fixed.

A group of proteins called ESCRTs is responsible for splitting the two cells apart if nothing is amiss. The abscission checkpoint blocks this process by altering certain proteins in the ESCRT complex, but exactly how this works is not yet clear.

To find out more, Strohacker et al. imaged ESCRT factors in a new experimental system in which the abscission checkpoint is active in many cells. This showed that, in this context, certain ESCRT components were rerouted from the thread of membrane between the daughter cells to previously unknown structures, which Strohacker et al. named abscission checkpoint bodies. These entities also sequestered other factors that participate in the abscission checkpoint and factors that contribute to gene expression.

These results are key to better understand how cells regulate their division; in particular, they provide a new framework to explore when this process goes wrong and contributes to cancer.

Once the abscission checkpoint is satisfied, cytokinetic abscission is mediated by the Endosomal Sorting Complexes Required for Transport (ESCRT) pathway, at least in transformed, cultured mammalian cells (*Carlton and Martin-Serrano, 2007*; *Gatta and Carlton, 2019*; *Morita et al., 2007*; *Vietri et al., 2020*). The ESCRT adaptor protein CEP55 provides an ESCRT recruiting platform (*Carlton and Martin-Serrano, 2007*; *Morita et al., 2007*; *Bastos and Barr, 2010*) at a protein-rich structure called the Flemming body (*Capalbo et al., 2019*; *Skop et al., 2004*) centrally located within the midbody. CEP55 forms rings on either side of the Flemming body and recruits the early-acting ESCRT factors, TSG101 (a component of the ESCRT-I complex) and ALIX, through a shared binding site (*Lee et al., 2008*). SEPT9 has also recently been implicated as a second, TSG101/ESCRT-I-specific midbody adaptor (*Karasmanis et al., 2019*). ALIX and TSG101/ESCRT-I, in turn, ultimately recruit late-acting ESCRT-III factors, including CHMP4B and IST1 through two parallel pathways (*Christ et al., 2016*). ESCRT-III subunits polymerize into membrane-constricting filaments and interact with the AAA-ATPase VPS4 to sever membranes in abscission zones on either side of the Flemming body (*Carlton and Martin-Serrano, 2007*; *Christ et al., 2016*; *Morita et al., 2007*). The abscission checkpoint must, therefore, inhibit ESCRT recruitment, polymerization and/or constriction to delay abscission.

Aurora B kinase (**AurB**) functions as the master regulator of the abscission checkpoint. AurB phosphorylates many different targets at the midbody, and thereby enforces abscission delay and stabilizes the midbody (*Hegemann et al., 2014*; *Kettenbach et al., 2011*; *Steigemann et al., 2009*). AurB is activated by phosphorylation (**pAurB**), both by the collective action of CLK1, 2, and 4 kinases and by autophosphorylation (*Petsalaki and Zachos, 2016*; *Yasui et al., 2004*). When the abscission checkpoint is satisfied, AurB is dephosphorylated and abscission proceeds (*Bhowmick et al., 2019*; *Capalbo et al., 2019*). Many pAurB targets likely remain to be identified, but one significant target is the regulatory ESCRT-III subunit, CHMP4C, a factor required for abscission checkpoint activity (*Capalbo et al., 2012*; *Carlton et al., 2012*). pAurB phosphorylates three sites within a region unique to CHMP4C that is required for the abscission checkpoint (*Figure 1A*; *Capalbo et al., 2012*; *Carlton et al., 2012*), and the ESCRT-binding kinase ULK3 phosphorylates other site(s) outside this insert region (*Caballe et al., 2015*). A naturally occurring variant allele of CHMP4C that reduces ALIX binding abrogates the abscission checkpoint, induces DNA damage accumulation, and correlates with increased susceptibility to several different cancers (*Pharoah et al., 2013*; *Sadler et al., 2018*). Thus, CHMP4C phosphorylation and its ALIX binding activity are necessary for abscission checkpoint function.

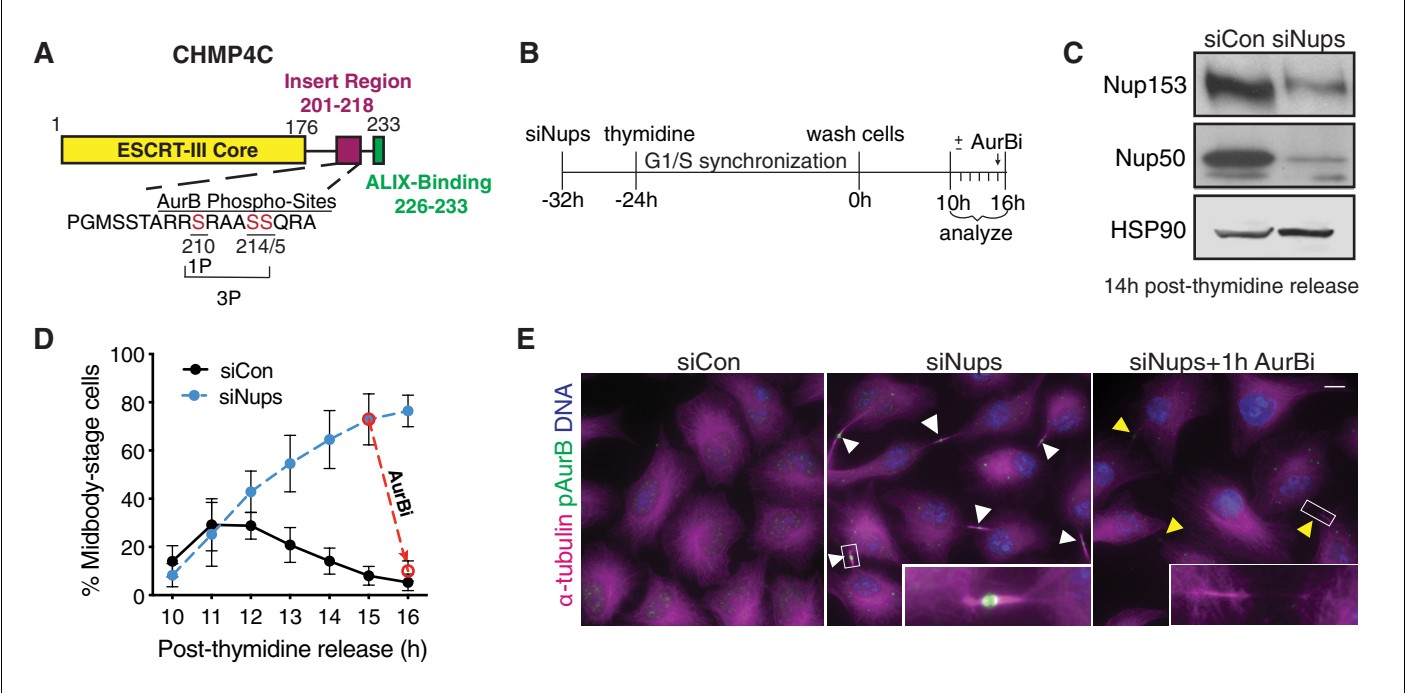

**Figure 1.** System for abscission checkpoint enrichment. (A) Schematic of the CHMP4C protein. (B) Timeline for synchronizing cells with an active abscission checkpoint using siNup153/siNup50 (siNups), followed by thymidine treatment. (C) Western blot of lysates prepared from control (siCon) and checkpoint-active (siNups) HeLa cells harvested 14 hr post-thymidine release. (D) Quantification of % midbody-stage cells after treatment as in (B). Dashed line shows the loss of midbody-stage cells from the siNups condition at t = 16 hr following addition of AurBi at t = 15 hr. N = 1200–3300 (total) cells/timepoint from n = 11 independent biological replicates. n = 5 biological replicates for siNups + AurBi. (E) Images acquired 16 hr post-thymidine release after treatment as in (B). White arrowheads: midbodies. Yellow arrowheads: recently abscised midbodies. Scale bar, 10 μm. Insets have enhanced brightness. <u>Throughout manuscript:</u> DNA is detected with DAPI unless noted. Bar and line graphs represent mean ± standard deviation. Times refer to timeline in (B) unless noted. N = total measurements from all biological replicates (n).

The online version of this article includes the following source data and figure supplement(s) for figure 1:

**Source data 1.** Source Data for *Figure 1*.

**Figure supplement 1.** Nup depletion and cell synchronization delay abscission without increasing chromatin bridges.

Checkpoint-induced abscission delay appears to be a multistep process, and previous studies have implicated: (1) phosphoregulation of several ESCRT-III activities (*Caballe et al., 2015*; *Capalbo et al., 2012*; *Carlton et al., 2012*), (2) actin patch formation to stabilize the intercellular bridge and prevent chromatin bridge breakage (*Bai et al., 2020*; *Dandoulaki et al., 2018*; *Steigemann et al., 2009*), and (3) sequestration of VPS4 in a single ring within the Flemming body, away from the abscission zones (*Thoresen et al., 2014*). In the latter case, phosphorylated CHMP4C acts together with the adaptor protein ANCHR to sequester VPS4 (*Thoresen et al., 2014*). Yet, the finding that CHMP4C must be able to bind ALIX for the checkpoint to function (*Sadler et al., 2018*) suggests additional crucial roles for CHMP4C. It remains unknown, however, where such interactions take place and whether cytoplasmic checkpoint regulatory mechanisms complement those at the midbody.

## Results

### Synchronization of cells with an active abscission checkpoint

The abscission checkpoint is normally active in only a small fraction of cultured cells because at any given time, few cells are in the cell cycle phase of cytokinesis and, further, because checkpoint activity is normally transient. We have overcome these experimental hurdles by combining treatments that (1) prevent checkpoint satisfaction (*Mackay et al., 2010*), using siRNA-mediated depletion of the nuclear pore basket proteins Nup153 and Nup50 (referred to throughout as siNups) and (2)

synchronize the cell cycle, using thymidine addition to arrest cells in G1/S phase, followed by release and synchronous progression through the remaining cell cycle (*Figure 1B,C*). These conditions elicited an active checkpoint in up to ~80% of HeLa cells in culture (*Figure 1D,E*). We did not observe increased chromatin bridges in these checkpoint-active samples (*Figure 1—figure supplement 1A, B*), indicating that lagging chromatin did not significantly contribute to sustained checkpoint activity. Importantly, cells enriched with an active abscission checkpoint by this method remained competent for abscission, as they all completed division within 60 min following deactivation of the checkpoint with an AurB inhibitor (**AurBi**, ZM 447439) (*Figure 1D,E*) or after >5 hr without treatment (not shown).

## Abscission checkpoint activity delays ALIX recruitment to the midbody

Using this new checkpoint enrichment protocol, we tested whether ESCRT factor recruitment to midbodies was altered by the abscission checkpoint. The ESCRT adaptor CEP55 and one of the two early-acting ESCRT factors, TSG101/ESCRT-I, localized normally to midbodies when the checkpoint was active (*Figure 2A,B*, *Figure 2—figure supplement 1A*, *Figure 2—figure supplement 2A,B*). In contrast, checkpoint activity significantly delayed midbody recruitment of the other early-acting ESCRT factor, ALIX. This delay was particularly pronounced in early-stage midbodies (*Figure 2C,D*) and was still striking when all midbodies were tracked over time (*Figure 2—figure supplement 1B*), despite a similar decrease in abundance of early-stage midbodies in both siCon and siNups samples (e.g., 15 hr post-thymidine, *Figure 2—figure supplement 1C*). To control for potential cell-to-cell variation, we also quantified CEP55 and ALIX intensity within the same cells to determine their relative ratios at individual midbodies. This experiment confirmed that an active checkpoint delays recruitment of ALIX to the midbody relative to CEP55, with an approximately threefold decrease in the relative ratio 11 hr post-thymidine release (*Figure 2E,F*, *Figure 2—figure supplement 1D*), and an approximately threefold decrease relative to TSG101/ESCRT-I in an analogous experiment (*Figure 2—figure supplement 2C,D*). By 16 hr after thymidine release, ALIX levels at the midbody had largely, but not completely recovered (*Figure 2D–F*, *Figure 2—figure supplements 1B,D* and *2D*). This is consistent with the duration of abscission delay under these conditions, which starts to lift after 16 hr (not shown). AurBi treatment rapidly rescued ALIX recruitment at 11 hr (*Figure 2G*), confirming that delays in ALIX recruitment were checkpoint dependent. We also found that checkpoint activity delayed recruitment of IST1, an ESCRT-III subunit that functions downstream of ALIX and is required for abscission (*Figure 2—figure supplement 3A,B*; *Agromayor et al., 2009*; *Bajorek et al., 2009*). IST1 recruitment was likewise restored by brief AurBi treatment (*Figure 2—figure supplement 3C*). Taken together, these observations reveal a new dimension of regulation that helps explain why abscission is delayed when this checkpoint is not satisfied. Both ALIX and IST1 localize to midbodies and are required for cytokinesis in HeLa cells (*Agromayor et al., 2009*; *Bajorek et al., 2009*; *Carlton and Martin-Serrano, 2007*; *Morita et al., 2007*), and abscission timing is therefore expected to be delayed when their recruitment is delayed.

## Abscission checkpoint bodies form in the cytoplasm when the abscission checkpoint is active

We next investigated what other cellular changes occur concomitantly with delayed ALIX recruitment. We have previously shown that cells expressing CHMP4C with mutations that inhibit ALIX binding bypass the abscission checkpoint (*Sadler et al., 2018*), indicating that interaction of CHMP4C and ALIX is key to checkpoint activity. We therefore first tested whether CHMP4C localization was also altered. Stably expressed HA-CHMP4C localized to midbodies regardless of checkpoint status, whereas ALIX recruitment was again delayed when the checkpoint was active (*Figure 3—figure supplement 1A–D*). Thus, consistent with previous reports (*Sadler et al., 2018*), bulk CHMP4C protein localization does not change with checkpoint activity. We next examined the localization of specific AurB-activated phospho-isoforms of endogenous CHMP4C using phospho-specific antibodies (*Capalbo et al., 2016*). As reported previously (*Capalbo et al., 2016*), singly phosphorylated CHMP4C (pCHMP4C) formed a single ring at the Flemming body and was recruited with the same timing whether or not the checkpoint was active (*Figure 3—figure supplement 1E–G*). Tri-phosphorylated CHMP4C (pppCHMP4C) also formed single Flemming body rings and, additionally, formed double rings, one on either side of the Flemming body (*Capalbo et al., 2016*),

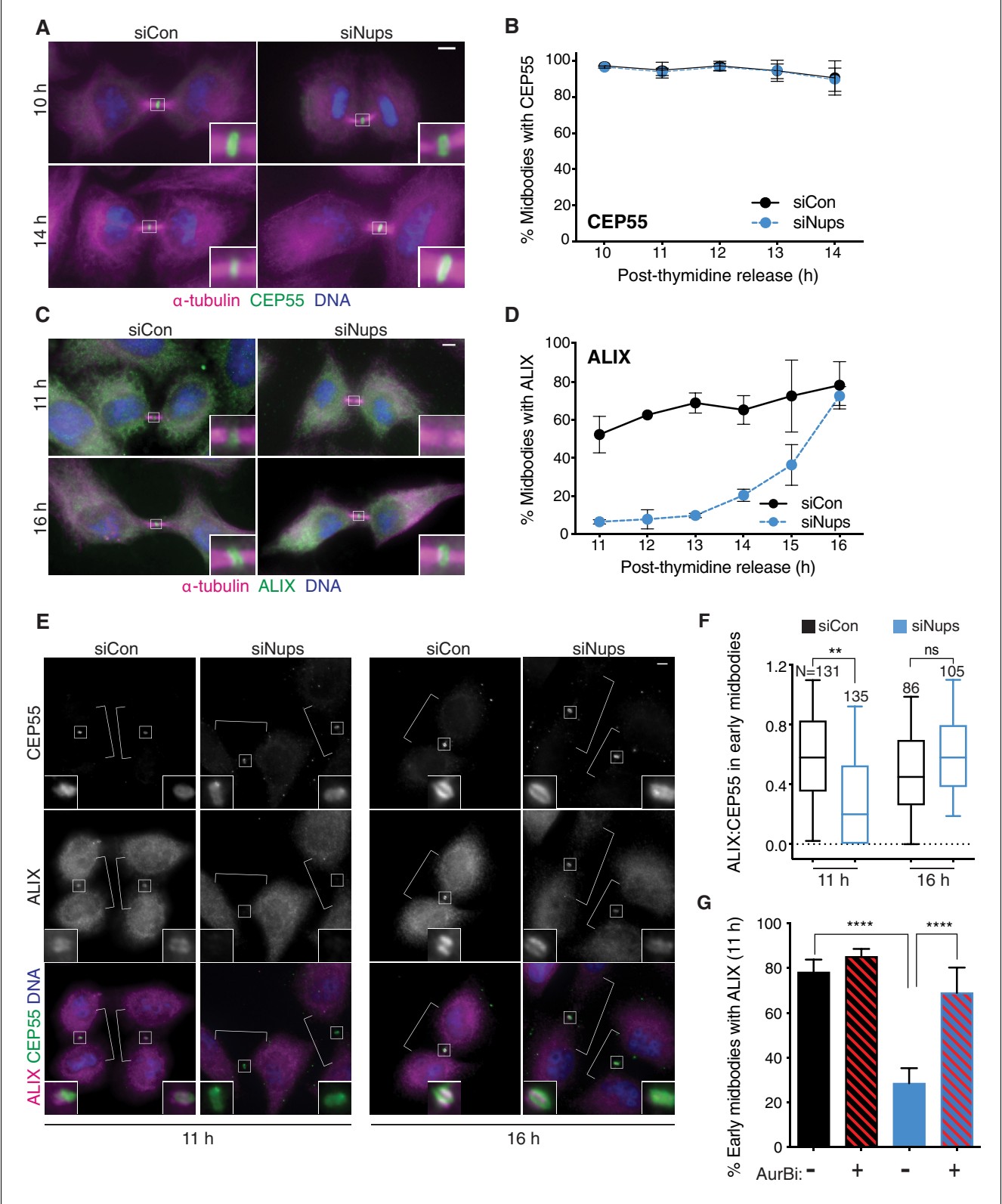

**Figure 2.** Abscission checkpoint activity delays ALIX recruitment to the midbody. Immunofluorescence and time course quantifications of CEP55 (A, B) and ALIX (C, D) recruitment to early-stage midbodies in control and checkpoint-active cells. N = 300 midbodies/timepoint from n = 3 biological replicates. (E, F) Immunofluorescence and quantification of ALIX:CEP55 relative intensities from n = 2 biological replicates in individual control and checkpoint-active early-stage midbodies (see *Figure 4B* for example of early midbody stage). Only one midbody is shown in the siCon/16 hr condition

*Figure 2 continued on next page*

*Figure 2 continued*

because this sample had fewer midbody-stage cells than other conditions. DNA detected with Hoechst. (**G**) Quantification of ALIX recruitment to early-stage midbodies at 11 hr with/without checkpoint enrichment (blue/black bars) and with/without AurBi added at 10.5 hr (striped/solid bars). N = 500 midbodies/treatment from n = 5 biological replicates. <u>Throughout manuscript</u>: scale bars are 5 μm unless noted. White brackets mark midbody-stage cells, as detected by α-tubulin, not shown. Data points without visible error bars (as in 2D siCon 12 hr) have a SD too small to display outside the data point or box. Boxplots represent the 25th, median, and 75th percentile values. Whiskers represent the 10th and 90th percentiles. $p<0.05$: *, $p\leq0.01$: **, $p\leq0.001$: ***, $p\leq0.0001$: ****. Exact p-values can be found in ***supplementary file 3***. See Materials and methods for statistical tests used.

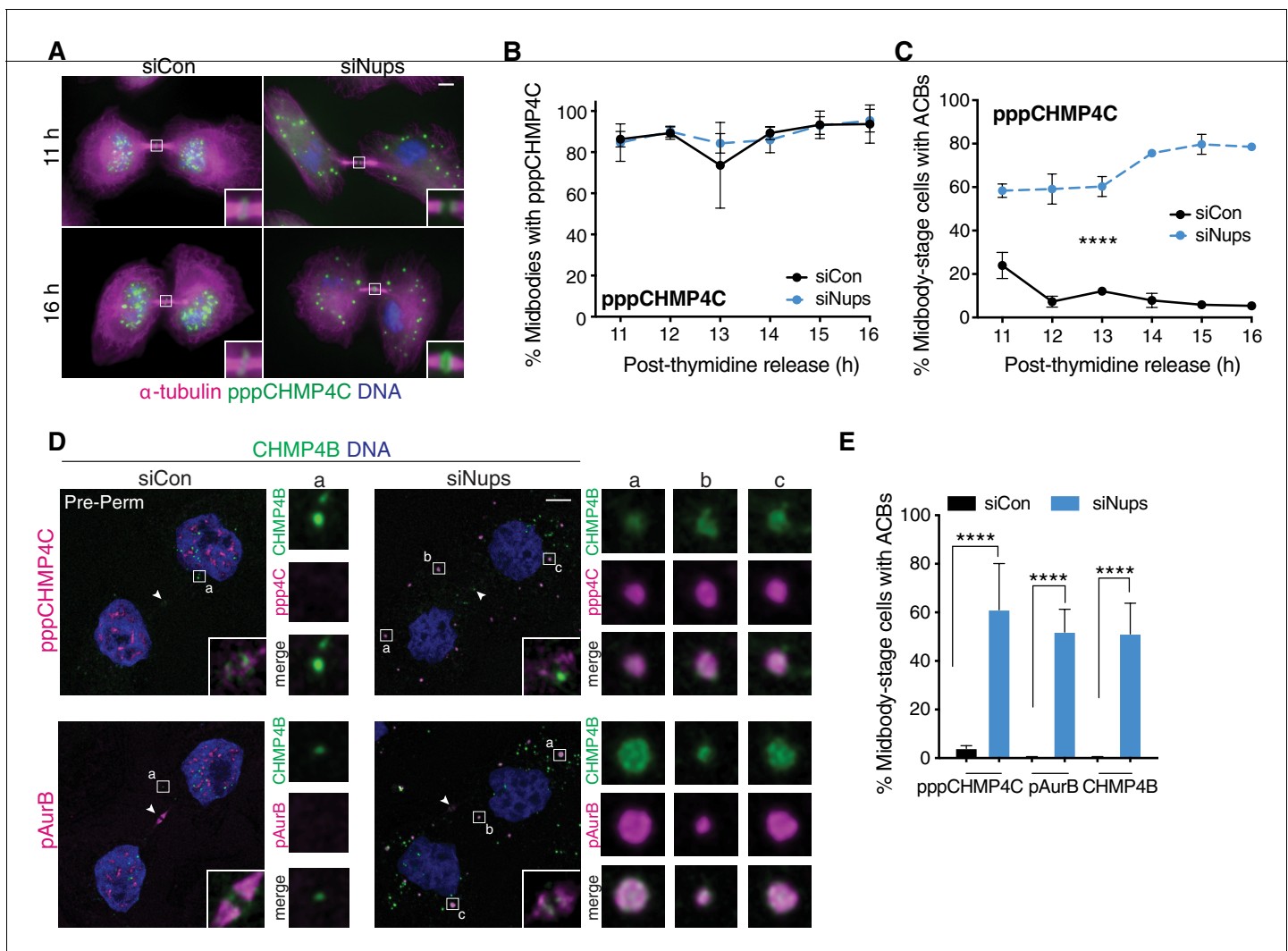

**Figure 3.** pppCHMP4C localizes to Abscission Checkpoint Bodies when the abscission checkpoint is active. (**A, B**) Immunofluorescence and time course quantification of pppCHMP4C recruitment to midbodies in control and checkpoint-active cells, and (**C**) time course quantification of midbody-stage cells with ACBs present. N ≥ 300 midbodies/timepoint from n = 3 biological replicates. (**D**) Confocal z-projections of pre-permeabilized midbody-stage cells under asynchronous conditions (48 hr after transfection with siNups or siControl), stained as indicated. Note the appearance of ACB component substructure in some cases (siNups, CHMP4B). (**E**) Quantification of midbody-stage cells with ACBs present (treated as in **D**). N = 300 midbodies/condition from n = 3 biological replicates. <u>Throughout manuscript</u>: Components are brighter in ACBs than in midbodies and the images are optimized for ACBs, which reduces the appearance of midbody localization in the confocal images. Confocal midbody insets are 2.3 μm wide and have enhanced brightness. White arrowheads mark Flemming bodies (where detectable). ACB enlargements are 2 μm wide.

The online version of this article includes the following source data and figure supplement(s) for figure 3:

**Source data 1.** Source Data for *Figure 3*.
**Figure supplement 1.** The abscission checkpoint does not delay CHMP4C recruitment to the midbody.
**Figure supplement 2.** CHMP4C is detected in ACBs, which are maintained in stable numbers when the abscission checkpoint is active.

particularly when the checkpoint was active (*Figure 3—figure supplement 1G*). However, checkpoint-dependent changes in pppCHMP4C distribution were much more dramatic at cytoplasmic sites (*Figure 3A,B*). In control midbody-stage cells, pppCHMP4C exhibited a granular nuclear localization, whereas cells with sustained checkpoint activity accumulated pppCHMP4C in 0.5–2 µm diameter cytoplasmic foci that we have named Abscission Checkpoint Bodies (**ACB**s). Pan-CHMP4C antibodies decorated ACBs, whereas antibodies specific to singly phosphorylated pCHMP4C did not, consistent with the tri-phosphorylated subpopulation of CHMP4C preferentially localizing to this site (*Figure 3—figure supplement 1E*, *Figure 3—figure supplement 2A*). Note, however, that detection of endogenous CHMP4C in ACBs with the pan-antibody required removing soluble cytoplasmic proteins by pre-fixation treatment with buffer (PHEM) to preserve general cell morphology followed by detergent to pre-permeabilize cell membranes. This treatment leaves larger cellular structures, including ACBs, intact while removing cytoplasmic background to increase ACB staining clarity. ACBs were observed in the majority of checkpoint-active cells throughout the entire post-thymidine time course (*Figure 3C*). ACBs were abundant (26 ± 14 per midbody-stage cell), and the number of ACBs per midbody-stage cell remained constant throughout the time course (*Figure 3—figure supplement 2B*). In summary, previously uncharacterized bodies that contain the abscission checkpoint factor pppCHMP4C form in the cytoplasm of cells in which the abscission checkpoint is active due to Nup153/50 depletion.

## CHMP4B and pAurB also localize to ACBs

Initial screens for additional ACB components revealed that a second CHMP4 isoform, CHMP4B, was also present at this site. These two CHMP4 isoforms have similar sequences, yet perform quite different roles in cell division: CHMP4C regulates abscission timing but is not required for abscission, whereas CHMP4B is required for abscission but does not have a regulatory role (*Capalbo et al., 2012*; *Carlton et al., 2012*; *Carlton et al., 2008*). CHMP4B is an abundant cytosolic ESCRT-III protein, and like bulk CHMP4C, its ACB localization was also best visualized after pre-permeabilizing cells to remove soluble cytoplasmic proteins (*Figure 3D*). An antibody specific for the T232 phosphorylated, active form of AurB (pAurB) (*Yasui et al., 2004*) also stained ACBs (*Figure 3D*), explaining our previous observation that abscission checkpoint activity caused pAurB to localize to cytoplasmic foci of unknown composition (*Mackay et al., 2010*). In the absence of checkpoint activation, CHMP4B exhibited a punctate cytoplasmic distribution in pre-permeabilized cells, but these puncta were significantly smaller than ACBs and did not colocalize with either pppCHMP4C or pAurB. Hence, in addition to pppCHMP4C, pAurB and CHMP4B are ACB components, and in each case, their ACB localization was strongly checkpoint dependent (*Figure 3E*).

## ACBs are derived from mitotic interchromatin granules

We further characterized ACBs by testing whether they corresponded to a variety of known cellular assemblies and organelles (*Figure 4—figure supplement 1A–C*). ACBs did not colocalize with a host of organelles, including P-bodies, which appeared the most similar in character and appearance yet were clearly distinct (*Figure 4—figure supplement 1D*). Unexpectedly, our survey revealed that ACBs are specifically detected with an antibody that recognizes SR domain-containing splicing factors as well as an antibody (mAb SC35) reported to primarily recognize the splicing factor SRRM2 (*Ilik et al., 2020*; *Figure 4A,B Figure 4—figure supplement 2A,B*). These antibodies are known to detect cytoplasmic bodies termed mitotic interchromatin granules (MIGs) (*Fu and Maniatis, 1990*; *Li and Bingham, 1991*; *Reuter et al., 1985*). During interphase, MIG components reside in a nuclear compartment called nuclear speckles where these components are hypothesized to be concentrated to increase splicing efficiency (*Beck, 1961*; *Chen and Belmont, 2019*). Both MIGs and nuclear speckles are compartments with liquid–liquid phase separation characteristics (*Rai et al., 2018*; *Strom and Brangwynne, 2019*), and they primarily contain factors that function in mRNA biogenesis, particularly splicing (*Mintz et al., 1999*; *Saitoh et al., 2004*; *Uversky, 2017*). After being released from nuclei upon mitotic nuclear envelope disassembly, these factors associate into small cytoplasmic MIG foci during the metaphase to anaphase transition. MIGs then normally disappear in late telophase, concomitant with stepwise reassembly of nuclear speckles within newly formed nuclei (*Prasanth et al., 2003*; *Reuter et al., 1985*; *Spector and Lamond, 2011*; *Spector and Smith, 1986*; *Tripathi and Parnaik, 2008*). In contrast, extended abscission checkpoint activity promoted the

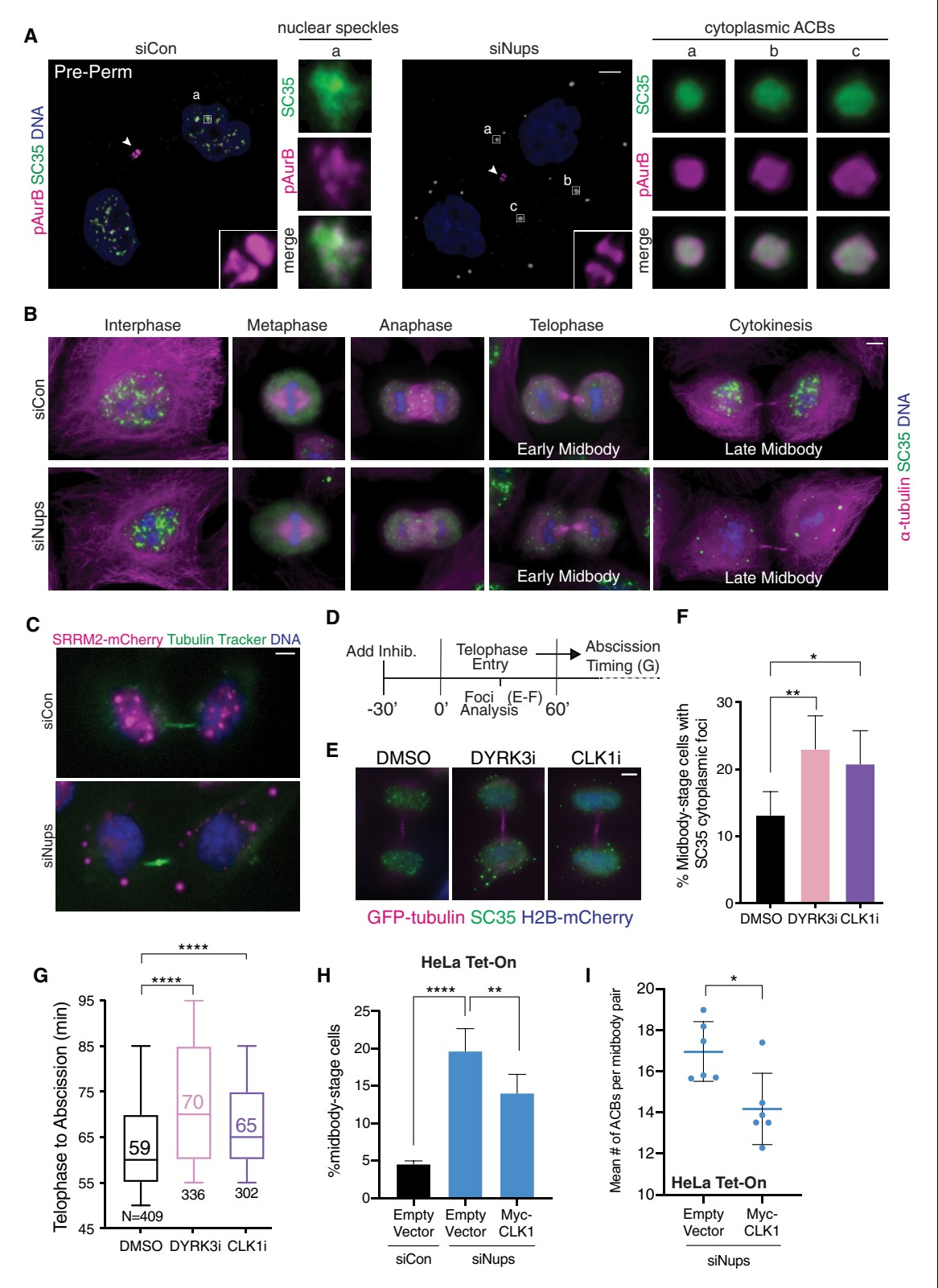

**Figure 4.** ACBs are related to MIGs and contribute to abscission delay. (**A**) Confocal z-projections of pre-permeabilized (Pre-Perm) midbody-stage cells showing that SC35 antibody and antibody against pAurB co-stain ACBs (asynchronous cultures, 48 hr after transfection with siNups or siControl). (**B**) Immunofluorescence of nuclear speckles, MIGs, and ACBs in asynchronous conditions (as in **A**, but non-pre-permeabilized and at a variety of cell-cycle stages as designated). (**C**) Live-imaging of HeLa cells expressing SRRM2-mCherry after 48 hr treatment with siCon or siNups. Images were

*Figure 4 continued on next page*

**Figure 4 continued**

independently adjusted for brightness and contrast for optimal display. (D) Timeline of live-cell imaging of abscission timing following treatment with DMSO, 1 µM DYRK3i, or 1 µM CLK1i. (E, F) Immunofluorescence of midbody-stage cells fixed 60′ after addition of vehicle or inhibitors, with quantification of % midbody-stage cells with cytoplasmic foci marked by SC35 antibody. N = 300 midbodies/condition from n = 3 biological replicates. (n = 4 for DMSO) (G) Timing from telophase to abscission (cells treated as in D). n = 3 biological replicates. (H) Quantification of midbody-stage cells after 72 hr treatment with siCon or siNups and 48 hr expression of empty vector or CLK1 kinase as indicated (asynchronous). N = 4800 cells from n = 6 biological replicates. (I) Mean number of ACBs per midbody pair after treatment as in (H). N ≥ 663 midbody-stage cells per condition. n = 6 biological replicates.

The online version of this article includes the following source data and figure supplement(s) for figure 4:

**Source data 1.** Source Data for *Figure 4*.
**Figure supplement 1.** ACBs do not colocalize with a variety of subcellular organelles and structures.
**Figure supplement 2.** Colocalization of ACB components.
**Figure supplement 3.** Interfering with timely resolution of MIGs delays cytokinetic abscission.
**Figure supplement 4.** CLK1 expression partially dissolves ACBs to mitigate abscission arrest.

appearance of ACBs, which are larger than MIGs and become apparent in late midbody-stage cells (*Figure 4B*). Nuclear speckle assembly in the nucleus was delayed concurrently with ACB appearance (*Figure 4A,B*). ACBs were not an artifact of fixation as we observed them in live cells expressing SRRM2-mCherry (*Figure 4C*). We therefore propose that cytokinetic ACBs are derived from MIGs and that abscission checkpoint signaling plays a pivotal role in inducing the transition.

When abscission checkpoint satisfaction occurred promptly, pAurB, pppCHMP4C, and CHMP4B, newly discovered here to be MIG constituents, relocalized individually to different sites in control cells at telophase, rather than transitioning to ACBs. Interestingly, the checkpoint maintenance factors pAurB and pppCHMP4C relocalized to nuclei where they partially colocalized with nuclear speckles (*Figure 4A*, *Figure 4—figure supplement 2A,B*). Nuclear speckle localization was not observed for CHMP4B, which remained cytoplasmic and also localized to midbodies (*Figure 3D*, *Figure 4—figure supplement 1B*; *Capalbo et al., 2012*; *Carlton et al., 2012*). Abscission checkpoint activity delayed timely nuclear relocalization of pAurB and pppCHMP4C, as well as splicing factors recognized by the canonical MIG marker mAb SC35, until abscission or just prior (*Figure 4A*, *Figure 4—figure supplement 2A,B*). These observations indicate that MIGs/ACBs play a role in constraining the migration of their constituents to various different cellular sites.

## ACBs regulate abscission timing

To probe for direct ACB functions in abscission delay, we tested whether abscission timing was altered when we artificially stimulated their formation/maintenance without otherwise preventing abscission checkpoint satisfaction. This was accomplished by inhibiting DYRK3, a kinase that prevents premature MIG formation in metaphase (*Rai et al., 2018*), or by inhibiting CLK1, a kinase that promotes component release from nuclear speckles and thus may similarly regulate MIG stability, allowing them to persist and at least partially mimic their maturation into ACBs (*Araki et al., 2015*; *Colwill et al., 1996*). CLK inhibition was previously reported to accelerate abscission (*Petsalaki and Zachos, 2016*), but our studies employed a CLK1/2-specific inhibitor (CLK1i) rather than a pan-CLK inhibitor, and our treatment windows were much shorter than those employed previously (30 min vs. 5 hr). For our studies, we treated with CLK1 inhibitor or DYRK3 inhibitor for 30 min to limit effects on preceding cell-cycle stages. Under these conditions, inhibition of either CLK1 or DYRK3 kinase increased the percentage of midbody-stage cells with SC35-positive cytoplasmic foci from 13% (control) to 23% (DYRK3i) or 21% (CLK1i) (*Figure 4D–F*) and increased median abscission times in the bulk population by 11 min (DYRK3i, 19% delay) or 6 min (CLK1i, 10% delay), as measured in live cell imaging experiments (*Figure 4G*). Similarly, the time course of midbody resolution in a synchronized cell population was delayed upon treatment with either inhibitor (*Figure 4—figure supplement 3*). Conversely, when we overexpressed CLK1 in a background of Nup153/50 depletion, midbody-stage cells decreased ~25% (*Figure 4H*). CLK1 overexpression did not completely dissolve ACBs, but did decrease the number of ACBs per midbody pair (*Figure 4I*, *Figure 4—figure supplement 4A*), and concomitantly increased the proportion of SRRM2 nuclear staining (*Figure 4—figure supplement 4B,C*), indicating that CLK1 overexpression induced SRRM2 relocalization from ACBs into nuclei. Thus, treatments that promote MIG/ACB formation and maintenance delay abscission in cells

independently of perturbations that prevent the abscission checkpoint from being satisfied, and treatments that promote ACB dissolution accelerate abscission, even when the abscission checkpoint is active. These effects are modest, but the experiments target just one aspect of abscission regulation in isolation and therefore support the model that ACBs have a functional role in abscission delay.

## ACB formation and ALIX recruitment delay occur in response to multiple mitotic errors and in different cell types

To confirm that ACBs are a broadly relevant component of the abscission checkpoint, we investigated their formation under the three other conditions known to prevent abscission checkpoint satisfaction: replication stress, intercellular tension, and lagging chromatin bridges within the midbody (*Lafaurie-Janvore et al., 2013*; *Mackay and Ullman, 2015*; *Steigemann et al., 2009*). We observed that ACB levels significantly increased in midbody-stage cells under conditions of replication stress (*Figure 5—figure supplement 1*) or high intercellular tension (*Figure 5—figure supplement 2*), but not in the presence of chromatin bridges (*Figure 5—figure supplement 3*). Delayed ALIX midbody recruitment followed this same trend (*Figure 5—figure supplements 1G*, *2D,* and *3D*). Thus, ACB formation and delayed recruitment of ALIX to the midbody were correlated and both were seen under three of four different conditions known to prevent satisfaction of the abscission checkpoint.

To test whether this newly identified checkpoint mechanism is deployed in other cellular contexts, we probed the non-transformed epithelial cell-line RPE1 for key checkpoint hallmarks. RPE1 cells exhibited abscission checkpoint activity in response to Nup depletion, as measured by an increase in midbody-stage cells and a delay in ALIX recruitment to the midbody (*Figure 5—figure supplement 4A–C*). This response was less robust than observed in HeLa cells, suggesting that corrective or compensatory pathways may mitigate errors that keep the abscission checkpoint from being satisfied or that RPE1 cells may be less dependent on the ESCRT pathway for abscission. Nonetheless, ACBs were once again detected in nearly all midbody-stage RPE1 cells subjected to the checkpoint enrichment protocol and contained pppCHMP4C, pAurB, CHMP4B, and splicing factors recognized by mAb SC35 (*Figure 5A–C*, *Figure 5—figure supplement 4D*). These observations demonstrate that checkpoint-dependent delay of ALIX midbody recruitment and ACB formation occur in distinct cell types and are broadly relevant as abscission checkpoint mechanisms.

## ACB formation requires CHMP4C

To probe the relationship between the abscission checkpoint and ACBs further, we depleted the key abscission checkpoint regulator CHMP4C and assayed ACBs. As expected, co-depletion of CHMP4C in Nup-depleted cells abrogated the checkpoint and reduced midbody-stage cells (*Figure 5D*, full bars). This treatment also significantly reduced the percentage of midbody-stage cells with ACBs (*Figure 5D*, shaded regions). The dependence of ACB formation on CHMP4C reinforces the connection between abscission checkpoint signaling and ACB appearance. The suppression of ACB formation in the absence of CHMP4C, despite Nup depletion, also demonstrates that ACBs do not form due to altered function of the nuclear pore per se (*Figure 5D*, *Figure 5—figure supplement 5*).

## ACBs contain ALIX, and ALIX depletion reduces ACB size

To investigate the connection between the presence of ACBs and the delay of ALIX recruitment to the midbody, we tested whether ALIX itself might target to ACBs. To avoid the strong signal for ALIX present in the cytoplasm (*Figure 2C*), we again used pre-permeabilization treatment to remove soluble cytoplasmic content. Using this method, we detected ALIX in ACBs in both HeLa and RPE1 cells (*Figure 6A,B*). To determine whether this localization was specific, we performed a co-depletion with siALIX and siNups, which was informative at several levels. First, as previously established, ALIX depletion alone resulted in cytokinesis failure (*Carlton and Martin-Serrano, 2007*; *Morita et al., 2007*; *Figure 6—figure supplement 1A,B*), and when abscission checkpoint satisfaction was prevented by concurrent treatment with siNups, enrichment of midbody-stage cells still took place (*Figure 6C*, full bars). This checkpoint activity corresponded to a significant reduction in cytokinesis failure (*Figure 6—figure supplement 1A–B*). These observations are consistent with the conclusion that reduced ALIX activity prevents abscission during checkpoint regulation. Second, depletion of ALIX confirmed its localization to ACBs as its detection at ACBs diminished

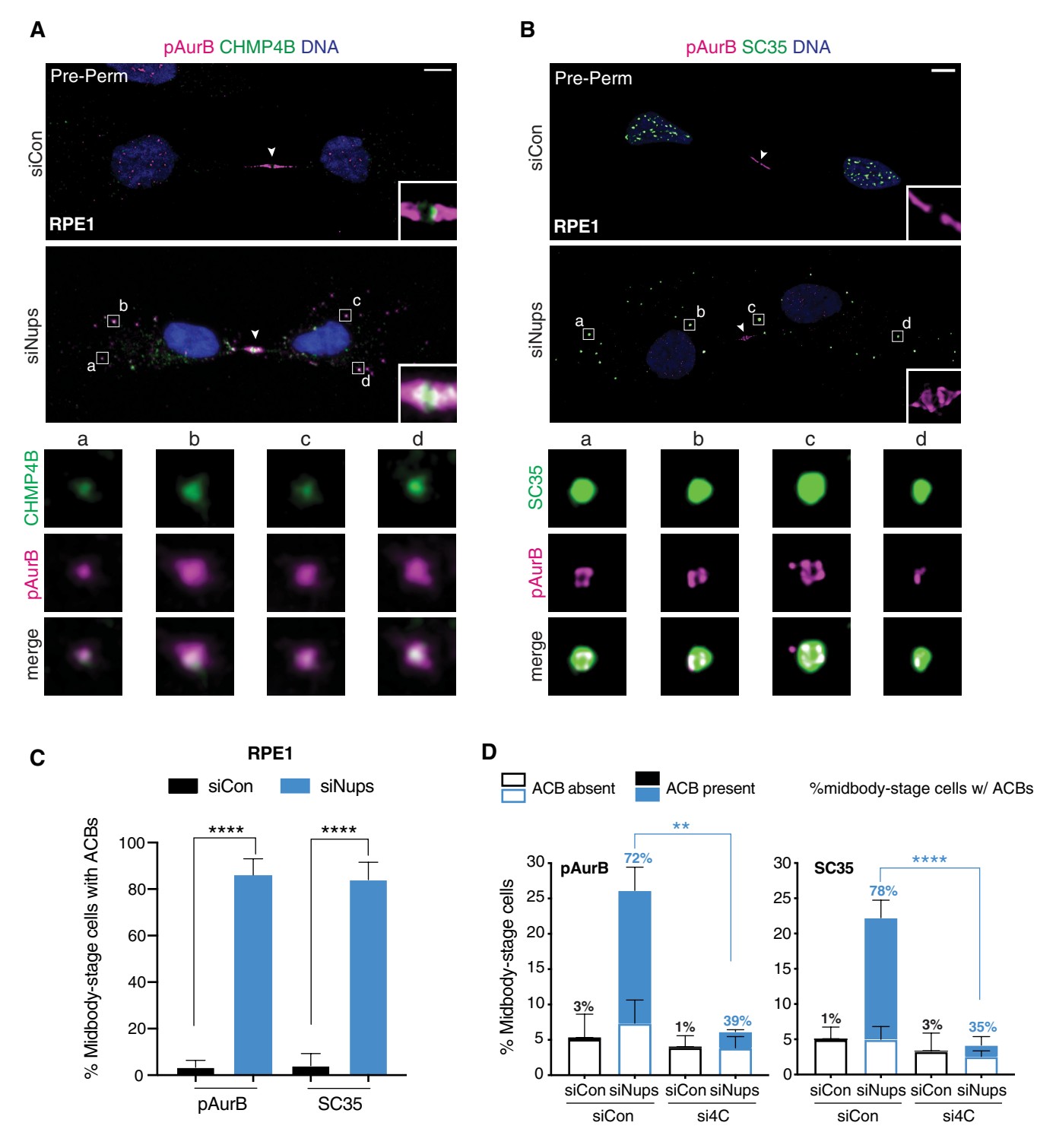

**Figure 5.** ACBs are conserved and dependent upon abscission checkpoint factor CHMP4C. (A, B) Confocal z-projections of pre-permeabilized control and checkpoint-active RPE1 cells following 14 hr thymidine release, stained for ACB markers as indicated. (C) Quantification of control and checkpoint-active RPE1 midbody-stage cells with ACBs present 14 hr post-thymidine release, detected by α-pAurB or SC35 antibody. N = 300 midbody-stage cells scored/condition from n = 3 biological replicates. (D) Quantification of midbody-stage HeLa cells under asynchronous conditions (72 hr after transfection with indicated siRNAs) containing ACBs (marked by α-pAurB or SC35 antibody). The number above each bar represents the % midbody-stage cells with ACBs present. N = 300 midbody-stage cells/condition from n = 3 biological replicates. p-values compare % midbodies with ACBs present.

*Figure 5 continued on next page*

*Figure 5 continued*

The online version of this article includes the following source data and figure supplement(s) for figure 5:

**Source data 1.** Source Data for *Figure 5*.
**Figure supplement 1.** ACB formation and ALIX recruitment delay occur following replication stress.
**Figure supplement 2.** ACB formation and ALIX recruitment delay occur when intercellular tension is heightened.
**Figure supplement 3.** Chromatin bridges do not significantly promote ACB formation or ALIX recruitment delay.
**Figure supplement 4.** RPE1 cells display hallmarks of the abscission checkpoint, including ALIX recruitment delay and ACB formation.
**Figure supplement 5.** Confirmation of efficiency and specificity of protein depletion when CHMP4C and Nup153/50 are simultaneously targeted.

concordantly (*Figure 6—figure supplement 2A,B*). Third, in the absence of ALIX, although ACB-like structures still persist in late midbody-stage cells in response to checkpoint activation (*Figure 6C*, shaded regions), the average cross-sectional area of these structures decreased by ~35% (corresponding to ~50% reduction in volume), as measured using two different ACB markers (*Figure 6D*, *Figure 6—figure supplement 1C*). Concomitantly, the number of the focal ACB-like structures increased in checkpoint-active midbody-stage cells depleted of ALIX (*Figure 6—figure supplement 1D*). Thus, ALIX is a component of ACBs and is required to create full-sized ACBs or to maintain their integrity. Interestingly, although pAurB and pppCHMP4C clearly localize to MIGs (*Figure 6—figure supplement 2E*), we could not detect ALIX in MIGs (*Figure 6—figure supplement 2C–D*). These observations indicate that ALIX is present in ACBs and, although we cannot rule out that our detection method is limiting for MIGs, it appears that ALIX recruitment contributes to ACB maturation from MIGs.

## ACB formation correlates with reduced midbody ALIX levels

The presence of ALIX in ACBs suggested that the checkpoint-dependent delay in ALIX recruitment to the midbody may occur because a modified subpopulation of ALIX that would normally be targeted to the midbody is instead retained in ACBs. To determine whether ACB formation and ALIX midbody recruitment delay are connected, cells were scored for both the presence of ACBs and the intensity of ALIX at the midbody, under conditions that enrich for abscission checkpoint-active cells. Consistent with a direct ALIX sequestration model, we found that the presence of ACBs strongly correlated with decreased ALIX levels at the midbody in individual cells (*Figure 6E*, *Figure 6—figure supplement 2F*). This correlation was striking even late in the post-thymidine time course, when overall ALIX midbody recruitment has begun to recover.

## Discussion

We have developed a new experimental strategy that is ideally suited for elucidating the regulation of abscission timing. Employing this assay, we found that abscission checkpoint activity can delay ALIX recruitment to the midbody and identified ACBs as a previously uncharacterized cytoplasmic body that contributes a new facet of abscission regulation. Specifically, we find that the abscission checkpoint restrains recruitment of ALIX and the downstream factor IST1 to the midbody, helping to explain how abscission is stalled in response to mitotic errors. Furthermore, we identified ACBs as a checkpoint-dependent compartment that concentrates splicing, checkpoint, and abscission factors, including mAb SC35-reactive splicing factors, pAurB, pppCHMP4C, CHMP4B, and ALIX. ACBs appear to be derived from MIGs, a compartment whose components and cell cycle behavior had previously been partially characterized, but whose functional role(s) is unknown at a mechanistic level (*Ferreira et al., 1994*; *Prasanth et al., 2003*; *Rai et al., 2018*; *Reuter et al., 1985*; *Spector and Lamond, 2011*; *Tripathi and Parnaik, 2008*; *Turner and Franchi, 1987*). ACBs also share similarities with cytoplasmic granules that have been observed in mouse testes (*Saitoh et al., 2012*). In this study, the presence of splicing factor-enriched cytoplasmic granules was found to correlate with decreased levels of particular nucleocytoplasmic transport factors. In light of our findings here, it will be of interest to resolve whether the cytoplasmic granules seen in spermatids are ACBs and, further, whether similar alteration of nucleocytoplasmic transport factors prevents abscission checkpoint satisfaction. Although much remains to be learned about MIG/ACB architecture and function, we found here that ACB formation/maintenance requires the abscission checkpoint factor CHMP4C and that the cytokinesis factor ALIX plays a distinct role in their maturation. CHMP4C and pAurB colocalize

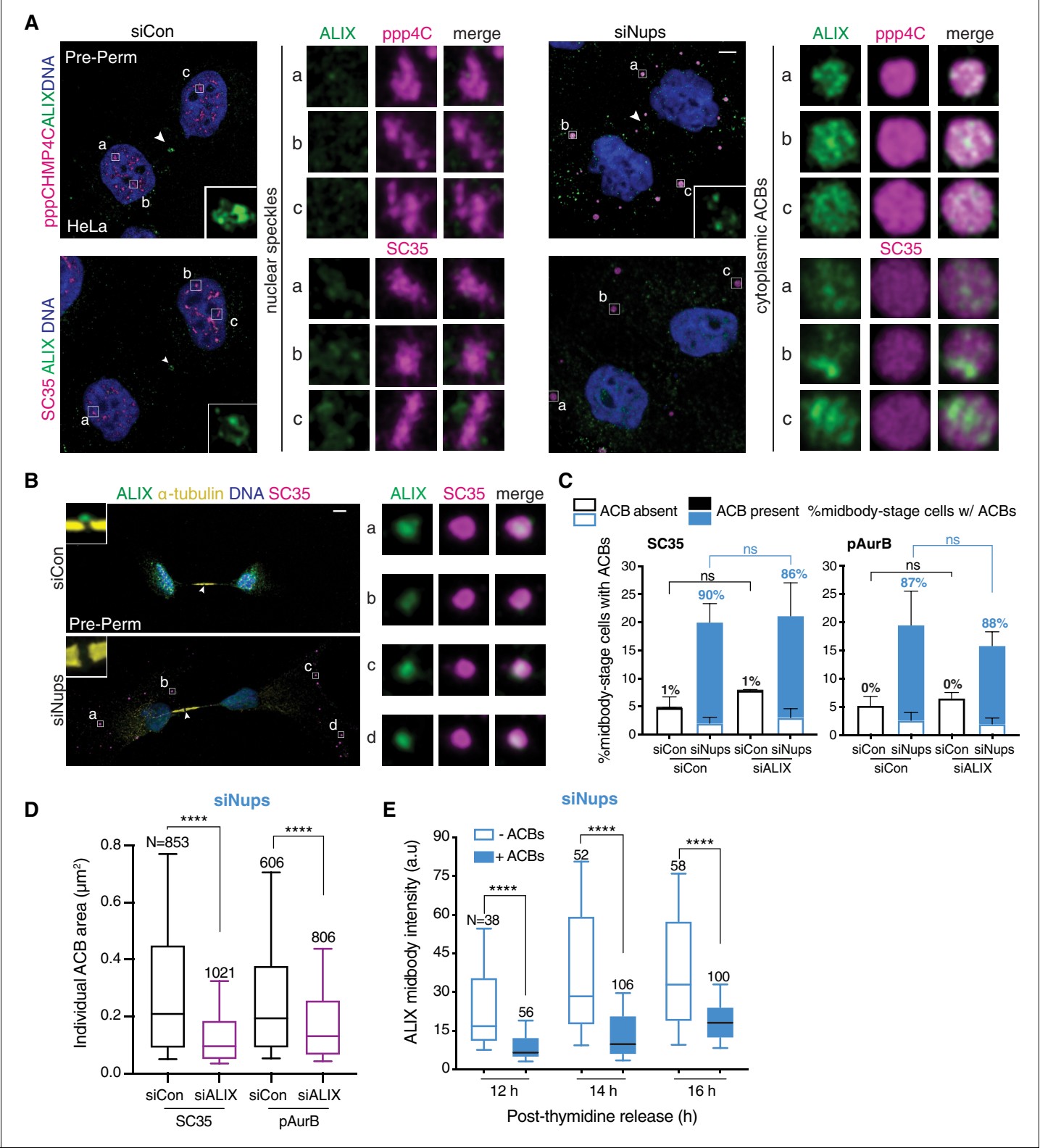

**Figure 6.** ACBs contain ALIX and cells with ACBs have delayed ALIX midbody recruitment. (**A**) Confocal z-projections of pre-permeabilized midbody-stage HeLa cells under asynchronous conditions (48 hr after transfection with siNups or siControl), stained as indicated. (**B**) Confocal z-projections of pre-permeabilized control and checkpoint-active midbody-stage RPE1 cells (14 hr post-thymidine release) stained as indicated. (**C, D**) Quantification of (**C**) midbody-stage cells with/without ACBs detected with α-pAurB or SC35 antibody, (**D**) ACB size under asynchronous conditions (72 hr after

*Figure 6 continued on next page*

*Figure 6 continued*

transfection with indicated siRNAs). (**C**) N = 300 midbodies scored/condition from n = 3 biological replicates. The number above each bar represents the % midbody-stage cells with ACBs present. p-values compare % midbodies with ACBs. (**D**) n = 4 biological replicates. (**E**) Time course quantification of relative ALIX midbody intensity in checkpoint-active cells. Midbody-stage cells were binned into categories with or without ACBs (marked by α-pAurB) using images from n = 3 biological replicates.

The online version of this article includes the following source data and figure supplement(s) for figure 6:

**Source data 1.** Source Data for *Figure 6*.
**Figure supplement 1.** ACB size but not abscission checkpoint arrest is dependent on ALIX.
**Figure supplement 2.** ALIX localizes specifically to ACBs.

within ACBs, which may favor formation and maintenance of the pppCHMP4C isoform because CHMP4C is an AurB substrate (*Capalbo et al., 2012*; *Carlton et al., 2012*). At least two ACB components, CHMP4C and CHMP4B, can bind ALIX (*McCullough et al., 2008*), and CHMP4C mutations that cripple or eliminate ALIX binding impair the checkpoint (*Sadler et al., 2018*), highlighting the functional importance of the CHMP4C–ALIX interaction in checkpoint maintenance, and raising the possibility that this interaction could target either the CHMP4 proteins or ALIX to ACBs.

In functional terms, targeting of ALIX to ACBs may sequester a particular subpopulation of this protein away from the midbody, contributing to the checkpoint-dependent abscission delay. Sequestration may serve to prevent both ALIX-mediated ESCRT-III recruitment to the midbody and stimulation of VPS4 activity, a role for ALIX that is suggested by recent results focused on its orthologue in yeast, Bro1 (*Tseng et al., 2021*). Previous studies have demonstrated that in order to bind CHMP4 proteins and function in abscission, ALIX must be activated by phosphorylation of two serine residues in its autoinhibitory C-terminal tail (*Sun et al., 2016*; *Zhai et al., 2011*). Phospho-activated ALIX is therefore an attractive candidate for the subpopulation of ALIX that is sequestered within ACBs. Interestingly, the distal end of this autoinhibitory tail also houses an intrinsically disordered region capable of higher order interactions that can form amyloids and viscous gels in vitro (*Elias et al., 2020*). This property may contribute to ALIX targeting to ACBs and/or its role in ACB growth.

The key role of ALIX in the abscission of cultured, transformed cells is now well established (*Carlton and Martin-Serrano, 2007*; *Christ et al., 2016*; *Morita et al., 2007*) and is reinforced by recent studies elucidating its stepwise recruitment to the abscission zone, where it works in a complex with syndecan4 and syntenin to recruit other ESCRT factors required for the scission event (*Addi et al., 2020*). However, mouse knockout models reveal a more complex picture in vivo. In these mouse models, ALIX and CEP55 were found to be required for normal brain and kidney development and cell division, whereas other organs appear largely unaffected (*Campos et al., 2016*; *Tedeschi et al., 2020*). Similarly, humans without functional ALIX display microcephaly and kidney defects but are otherwise healthy and can live into their 20's (*Khan et al., 2020*). We reconcile these observations with our model of the abscission checkpoint in the following ways: (1) Redundant pathways may ensure that abscission (and an abscission checkpoint) takes place in vivo; indeed, a recent report argues that ESCRT proteins, including ALIX, are still recruited to the midbody in the CEP55 knockout mouse (*Little et al., 2021*), and in cases where ALIX is absent, TSG101 may dominate in driving ESCRT activity (*Christ et al., 2016*; *Karasmanis et al., 2019*). (2) It is also possible that tumor-derived cells are more dependent on the ESCRT pathway for abscission than non-cancerous cells, albeit with the exception of the developing brain and kidney. In line with this, ESCRT factors are often overexpressed in cancer (*Jeffery et al., 2016*; *Lin et al., 2020*), and our results show a more robust abscission checkpoint response in HeLa cells compared to non-transformed RPE1 cells. This important issue requires further study, but could potentially be used to advantage in therapeutic approaches.

ALIX targeting to ACBs is the first known cytoplasmic mechanism for abscission checkpoint regulation, but several lines of evidence indicate that this is just one of a repertoire of mechanisms that can be integrated to accomplish abscission delay. For example, the ACB mechanism does not function measurably when the checkpoint is induced by lagging chromatin bridges (*Figure 5—figure supplement 3*). Consistent with that observation, ALIX does not appear delayed in its recruitment to the midbody under these circumstances. Thus, the chromosomal bridge structure appears to connect to AurB-dependent abscission delay via distinct mechanisms. Indeed, a histone

acetyltransferase complex plays an integral role in response to chromatin in the cell cleavage plane in the analogous NoCut checkpoint in yeast (*Mendoza et al., 2009*), and condensin has been proposed to contribute to abscission delay in the presence of chromatin bridges in *C. elegans* (*Bembenek et al., 2013*). There is also other precedent for modularity in abscission checkpoint mechanisms elicited by different cues. For example, although phosphorylation of IST1 by ULK3 is required for an abscission delay in response to chromatin bridges or Nup depletion, it does not appear to be required for tension-mediated abscission regulation (*Caballe et al., 2015*). Furthermore, other abscission delay mechanisms, such as ANCHR-dependent sequestration of the ATPase VPS4 away from the abscission zone (*Thoresen et al., 2014*), may be deployed in particular combinations depending on the error present.

Finally, our demonstration that late midbody-stage ACBs are derived from telophase MIGs begs the question of why these bodies colocalize factors that function in both abscission and mRNA biogenesis. Interphase nuclear speckles are hypothesized to be transcription hubs that mediate efficient splicing especially of active genes, and molecular detail of their function is emerging (*Smith et al., 2020*). The functional roles of MIGs are currently less clear. Abnormal MIG assembly triggers metaphase arrest (*Rai et al., 2018*; *Sharma et al., 2010*), implying a role in mitotic progression, but the mechanism is not yet understood. We have shown that artificially promoting ACB assembly triggers an abscission delay (*Figure 4D–G*, *Figure 4—figure supplement 3*) and that reducing ACB formation accelerates abscission (*Figure 4H,I*, *Figure 4—figure supplement 4*), indicating that ACBs have a functional role in cytokinetic progression. Intriguingly, our observations suggest that ACBs are normally remodeled and nuclear speckle reformation initiated before abscission takes place. It will therefore be of interest to determine whether sequestration and coordinated release of the splicing factors or other regulatory factors present within ACBs are required for abscission regulation and cytokinetic progression.

# Materials and methods

**Key resources table**

| Reagent type (species) or resource | Designation | Source or reference | Identifiers | Additional information |
|---|---|---|---|---|
| Cell line (*Homo sapiens*) | HeLa-N | Maureen Powers Lab | | HeLa cells selected for transfectability, ID confirmed by STR profiling |
| Cell line (*Homo sapiens*) | RPE1 | Bruce Edgar Lab | RRID:CVCL_4388 | Non-transformed, ID confirmed by STR profiling |
| Transfected construct (*Homo sapiens*) | pLVX-CLK1 | This paper | Addgene Cat#:174088 | Lentiviral construct to inducibly express CLK1 |
| Antibody | anti-ALIX (Rabbit polyclonal) | Covance (This Lab) | RRID:AB_2892637 | IF (1:500), WB (1:500) |
| Antibody | anti-pAurB (Rabbit polyclonal) | Rockland | Cat#: 600-401-677 RRID:AB_206164 | IF (1:500) |
| Antibody | anti-pppCHMP4C (Rabbit polyclonal) | Pier Paolo D'Avino | N/A | IF (1:500) |
| Antibody | anti-SC35 (mouse monoclonal) | Abcam | Cat#: ab18826 RRID:AB_298608 | IF (1:5000) |
| Antibody | anti-SRRM2 (rabbit polyclonal) | Thermofisher | Cat#: PA5-66827 RRID:AB_2665182 | IF (1:1000) |
| Sequence-based reagent | siNup153 | *Mackay et al., 2010* | siRNA | GGACUUGUUAGAUCUAGUU |
| Sequence-based reagent | siNup50 | *Mackay et al., 2010* | siRNA | GGAGGACGCUUUUCUGGAU |
| Chemical compound, drug | CLK1/2 Inhibitor | Millipore | 534350 | 1 µM |

*Continued on next page*

*Continued*

| Reagent type (species) or resource | Designation | Source or reference | Identifiers | Additional information |
|---|---|---|---|---|
| Chemical compound, drug | DYRK3 Inhibitor | Tocris | GSK 626616 | 1 µM |
| Chemical compound, drug | AurB Inhibitor | Bio-Techne | ZM 447439 | 2 µM |
| Chemical compound, drug | Thymidine | CalBiochem | CAS 50-89-5 | 2 mM |
| Software, algorithm | Fiji | NIH | RRID:SCR_002285 | |

### Antibodies

Details regarding antibodies used in this study can be found in *Supplementary file 1*.

### Plasmids

Details regarding plasmids used in this study can be found in *Supplementary file 2*. DNA was amplified using PCR and ligated into the pLVX-inducible vector using Gibson Assembly according to the manufacturer's instructions (NEB, Rowley, MA).

### Cell culture

HeLa cells were cultured and maintained at 37°C and 5% $CO_2$ in DMEM supplemented with 10% FBS. The Tet-On HeLa dox-inducible and HA-CHMP4C (*Carlton et al., 2012*) cell lines were supplemented with 100 µg/ml G418 to maintain Tet-On or HA-CHMP4C expression respectively (Invitrogen, Carlsbad, CA). RPE1 cells were supplemented with 10 µg/ml hygromycin (Invitrogen) to maintain hTERT expression. The SRRM2-mCherry cell line was maintained in 100 µg/ml G418 to maintain Tet-On expression and 10 µg/ml puromycin (InvivoGen, San Diego, CA) for SRRM2-mCherry expression. At the outset of these studies, HeLa, HeLa Tet-On, and RPE1 cells were tested and found negative for mycoplasma using a PCR mycoplasma detection kit (ABM, Bellingham, WA). Cell types were authenticated by sequencing 24 loci (University of Utah Sequencing Core). Unless otherwise labeled in individual panels or figure legends, all experiments used HeLa cells.

### Cell lines

To generate a stable cell line, Day 1: Hela Tet-On cells were plated in a 24-well dish. Day 2: cells were transfected with 500 ng/well of the SRRM2-mCherry plasmid using Lipofectamine LTX with Plus Reagent according to the manufacturer's instructions (Thermo Fisher Scientific, Waltham, MA). Day 3: cells were split into a 10 cm dish with 500 µg/ml G418 and 10 µg/ml Puromycin for selection. After drug selection for 14 days, colonies were harvested into individual wells of 12-well plates. Clones were validated for inducible SRRM2-mCherry expression using immunofluorescence imaging and an antibody specific to mCherry (Abcam, Cambridge, UK, ab167453).

### siRNA transfections

Cells were singly transfected with siRNA for 42–72 hr, as indicated in figure legends, using Lipofectamine RNAiMax (ThermoFisher) according to the manufacturer's instructions. Media was exchanged 24 hr after transfection, and cells were incubated between 18 and 48 hr before harvesting. For co-transfections, cells were seeded with combinations of (1) siALIX, siCon, or siNup153/50, or (2) siCHMP4C, siCon, or siNup153/50. Media was exchanged 24 hr after transfection. Cells were fixed 72 hr post-transfection to allow adequate time for CHMP4C or ALIX knockdown. Details regarding siRNAs used in this study are provided in *Supplementary file 2*.

### Immunofluorescence

Cells were seeded on acid-washed, 10 µg/ml fibronectin-treated, glass coverslips and then treated according to the individual experimental protocol. To fix, cells were removed from the incubator, washed once in PBS pH 7.4, and then fixed by one of the three methods as notated in *Supplementary file 1*:

1. 10 min, −20℃ methanol,
2. 15 min, 4℃, 2–4% paraformaldehyde (PFA) followed by 5 min, 23℃, 0.5% Triton-X in PBS,
3. rinsed with PHEM buffer at 23℃ (50 mM PIPES, 25 mM HEPES pH 7.0, 10 mM EGTA, 4 mM MgSO$_4$, with PMSF [1 mM], aprotinin, leupeptin added freshly), pre-permeabilized with 0.5% Triton-X in PBS for 1 min at 23℃, fixed with 2–4% ice cold PFA for 15 min, and incubated with 0.5% Triton-X in PBS for 5 min at 23℃.

After fixing, cells were rinsed twice with 2 ml PBS each time and then blocked (3% FBS and 0.1% Triton X-100 in PBS) for 30 min on the bench top. Primary antibodies were applied at the dilution noted in *Supplementary file 1* for at least 1 hr, 23℃ in blocking solution. After 1 wash with 2 ml PBS, secondary antibodies (Thermofisher) were applied for 45 min to 1 hr, and cells were washed in 2 ml PBS. For *Figure 2E*, *Figure 2—figure supplement 2C*, DNA was stained with Hoechst for 10 min at 23℃. For all other images, coverslips were mounted with ProLong Gold Antifade Reagent with DAPI (Thermofisher) on a microscope slide.

## Imaging

Images were acquired using four different microscopes:

1. Leica SP8 Confocal 63× 1.4 oil HC PL APO objective with adjustable white-light laser to control for bleed-through (*Figure 3D*, *Figure 4A*, *Figure 5A,B*, *Figure 6A,B*, *Figure 3—figure supplement 2A*, *Figure 4—figure supplement 1C*, *Figure 4—figure supplement 2A,B*, *Figure 6—figure supplement 1C*, *Figure 6—figure supplement 2A,C*). Images were acquired as z-stacks and each individual slice was deconvolved using the Hyvolution and Lightening modes on Leica App Suite X Software. Presented images are maximum z-projections of the deconvolved slices.
2. Zeiss Axioskop 2 Widefield 63× 1.4 oil DIC Plan Apochromat objective (*Figure 1E*, *Figure 2A,C*, *Figure 3A*, *Figure 4B*, *Figure 2—figure supplement 2A*, *Figure 2—figure supplement 3A*, *Figure 3—figure supplement 1B,E,G*, *Figure 4—figure supplement 1A,B,D*, *Figure 4—figure supplement 3E*, *Figure 4—figure supplement 4C*, *Figure 5—figure supplement 1D, F*, *Figure 5—figure supplement 2C,E*, *Figure 5—figure supplement 4D*, *Figure 6—figure supplement 1A*, and *Figure 6—figure supplement 2E,F*). Images were acquired as single-plane widefield images.
3. Axioskop two mot PLUS, 63× 1.4 oil and 100× oil Plan-Apochromat objectives (*Figure 2E*, *Figure 4E*, *Figure 2—figure supplement 2C*, and *Figure 5—figure supplement 3A,C*). These images were acquired as single-plane widefield images.
4. Nikon Ti-E widefield inverted microscope (Nikon 60x N.A. objective lens) and Andor Zyla CMOS camera (Andor, Manchester, CT) (*Figure 4C*).

## Live imaging of ACBs

The SRRM2-mCherry-inducible cell line was plated in four-well LabTek chamber slides. Day 1: cells were treated with 2 µg/ml Doxycycline to induce SRRM2-mCherry expression. Day 2: cells were treated with siCon or siNups as above. Day 4: cells were treated with one drop NucBlue reagent (ThermoFisher) per 2 ml of imaging media and 1 µM Tubulin Tracker Green (Thermofisher) 45 min prior to imaging in HEPES buffered, Phenol Red-free DMEM/F-12 (ThermoFisher) containing 10% FBS. Cells were imaged on a Leica SP8 White Light Nikon Ti-E widefield inverted microscope at 60× objective.

## Cell and image scoring

Midbody-stage cells were identified by α-tubulin staining and were always counted as one cell. Cell populations were counted and sorted into interphase, midbody-stage, mitotic, recently abscised pair (counted as one cell), multinucleate, and failed bridge (counted as one cell). Cells were counted unblinded while on the microscope.

Midbodies were designated as 'early' or 'not early' using α-tubulin staining before scoring for the examined phenotype. Midbodies were classified as early based on midbody width, level of midbody-pinching, nuclear area, and flatness of cells, see *Figure 4B* for illustration.

Midbodies were scored as having ACBs if they had at least one ACB. However, most midbody-stage cells with ACBs had 10–40 ACBs. For most figures, cells were counted as having or not having

ACBs, unblinded while on the microscope. In *Figure 3—figure supplement 2B* and *Figure 4I*, ACBs were counted from images using 'Find Maxima' with Fiji software (NIH). To quantify ALIX signal in ACBs, deconvolved z-slices were stacked and then uniformly adjusted for brightness and contrast. The SC35 or pAurB channels were uniformly thresholded, and then cytoplasmic objects between 0.025 and 5 $\mu m^2$ were marked as regions of interest (ROIs). ROIs were used to measure individual ACB intensity and area in the non-thresholded ALIX channel. Values were background corrected with large ROIs in the cytoplasm that excluded ACBs.

Quantification of fluorescence staining intensity at the midbody was also done with Fiji software (NIH). The freehand selection tool was used to outline the region of interest at the midbody, and staining intensity within this area was measured. Signals were background corrected using measurements from adjacent regions. In cases where the protein was not visible at the midbody, these methods sometimes generated a small negative intensity value after background correction because the Flemming body is a natural dark zone. In these cases, intensity was valued at zero.

To quantify SRRM2 subcellular distribution, the DAPI channel was used to threshold the nuclei and create ROIs using Fiji software. These ROIs were then used to determine the mean SRRM2 nuclear intensity. For each midbody-stage cell, an N:C ratio was determined by dividing the average mean nuclear intensity by the baseline cytoplasmic signal (the mean cytoplasmic intensity measured in an area devoid of ACBs).

## Immunoblotting

Cells were lysed in NP40 lysis buffer (50 mM Tris pH 7.4, 250 mM NaCl, 5 mM EDTA, 50 mM NaF, 1 mM $Na_2VO_4$, 1% Nonidet P40, 0.02% $NaN_3$, with freshly added PMSF [1 mM], aprotinin, leupeptin) for 30 min, 4°C, vortexing every 10 min. Lysates were clarified by spinning at 13,200 g for 10 min at 4°C, and protein contents in clarified lysates were quantified by Bradford assay before gel loading. Gel and western conditions are listed in *Supplementary file 1*. Twenty to 35 µg lysate per sample was prepared with SDS loading buffer, resolved by SDS–PAGE, and following wet transfer for 2 hr at 40 V, membranes were blocked with 5% milk in Tris-buffered saline (TBS) for at least 30 min, 23°C, and were incubated with primary antibodies in 5% milk in TBS-T (0.1% Tween 20/TBS) overnight at 4°C. Membranes were washed in TBS-T and incubated with the corresponding secondary antibodies conjugated with HRP (Thermofisher) or near-infrared fluorescent dyes (Abcam, Cambridge, UK) in TBS-T for 1 hr, 23°C, and washed again. Blots were detected with Western Lightning PLUS ECL (PerkinElmer, Waltham, MA) on Hyblot CL film (Thomas Scientific, Swedesboro, NJ). Proteins with infrared fluorescent dyes were detected using a Li-Cor Odyssey Infrared scanner and Image Studio version 5.2 software.

## Checkpoint activation

The abscission checkpoint was kept active using one of four different methods.

1. Nup depletion: cells were seeded on glass coverslips and transfected with 10 nM siNup153 and 10 nM siNup50 or control siRNA. In asynchronous experiments, media was exchanged at 24 hr, and cells were fixed at 48 hr or 72 hr. In synchronous experiments, 2 mM thymidine (Calbiochem, San Diego, CA) was added to samples 8 hr post-seeding. Cells were incubated with thymidine for 24 hr and then washed with PBS three times to remove thymidine and siRNA transfection mixture. Fresh media was added, and cells were harvested 10–18 hr later.
   In cases where cells were treated with AurB inhibitor (AurBi), 2 µm ZM 447439 (*Ditchfield et al., 2003*) (Bio-Techne, Minneapolis, MN) or DMSO vehicle was added 30 min (for partial abscission completion in *Figure 2G*, *Figure 2—figure supplement 3C*) or 1 hr (for almost complete abscission completion in *Figure 1D,E*) prior to fixing.
2. Chromatin bridges: cells were seeded on acid-washed glass coverslips and incubated for 48 hr without perturbation. Cells were fixed and stained with Lap2ß to identify the small percentage of cells with naturally occurring chromatin bridges, with or without accompanying tubulin staining. Phenotypes were scored in cells with and without chromatin bridges.
3. Replication stress: following the general procedure previously described (*Mackay and Ullman, 2015*), cells were seeded on glass coverslips, and 2 mM thymidine was added 8 hr post-seeding to synchronize cells. Cells were incubated for 24 hr, and cells were washed three times with 2 ml PBS to remove thymidine. Fresh media was added and supplemented with 0.4 µM aphidicolin or DMSO. Cells were harvested 12–18 hr later. In *Figure 5—figure supplement*

**1C**, 0.4 µM aphidicolin was added to cells at the time of seeding. Cells were incubated 48 hr and then harvested.

4. Tension: following the general procedure previously described (*Lafaurie-Janvore et al., 2013*), in which intercellular tension is limited by cell density, cells were seeded on glass coverslips at a density of 25,000 cells/ml (low density, high tension) or 75,000 cells/ml (high density, low tension). Cells were harvested at 48 hr post-seeding without additional perturbations. Cells were only scored as 'high tension' if at least one side was not touching a neighbor and was free to expand and scored as 'low tension' if completely surrounded by other cells and not free to spread.

## ACB induction experiments

For fixed-imaging time courses, HeLa cells were seeded on glass coverslips in 24-well dishes, and 2 mM thymidine was added 8 hr post-seeding. After 24 hr, thymidine was washed out with PBS. Thirteen hours post-thymidine release, after most cells had completed metaphase, 1 µM DYRK3 inhibitor (GSK 626616, Tocris, Minneapolis, MN), 1 µM CLK1/2 inhibitor (534350, Millipore, Burlington, MA), or DMSO were each added individually to six separate wells. Coverslips from wells corresponding to each treatment were fixed directly after inhibitor/vehicle was added for the first timepoint at 13 hr post-thymidine release and then at hourly intervals until 18 hr post-thymidine release.

Live-imaging experiments to determine abscission timing used HeLa cells expressing H2B-mCherry and GFP-α-tubulin. Cells were seeded in a Lab-Tek II 8-chambered #1.5 German Coverglass System and incubated for 48 hr. Stage positions were first set on the microscope, and then 1 µM DYRK3i, 1 µM CLK1/2i, or DMSO were added just prior to initiation of imaging. Imaging was carried out for 3 hr on a Nikon Ti-E widefield inverted microscope (Nikon 20x N.A. dry objective lens) equipped with Perfect Focus system and housed in a 37°C chamber (OKOLAB, Ambridge, PA) with 5% $CO_2$. Multiple fields of view were selected at various x and y coordinates, and images were acquired using a high-sensitivity Andor Zyla CMOS camera (Andor, Manchester, CT) controlled by NIS-Elements software. Images were acquired every 5 min, and abscission time was measured as the time from midbody formation to disappearance. In parallel, cells were seeded on glass coverslips, incubated for 48 hr, and either DYRK3i, CLK1i, or DMSO were added for 60 min prior to fixing and detection of foci labeled with mAb SC35 or antibodies against pAurB.

## CLK1 expression experiments

On Day 1, cells were seeded into 24-well dishes both with and without glass coverslips and transfected with either siCon or siNups siRNA as described above. On Day 2, cells were transfected with 800 ng/well of empty vector DNA or Myc-CLK1-WT DNA using Lipofectamine LTX according to the manufacturer's instructions (Thermofisher). At the time of transfection, 2 µg/ml doxycycline was added to the medium to induce expression. On Day 3, media was changed and fresh media added with 2 µg/ml doxycycline. On Day 4, cells were harvested for immunofluorescence and immunoblot as described above.

## Statistical analysis

Five statistical tests were used to evaluate the significance of data. When comparing two samples, p-values were calculated using an unpaired t-test or Mann–Whitney test for non-normally distributed datasets (*Figure 4I*, *Figure 1—figure supplement 1A*, *Figure 4—figure supplement 4B*, *Figure 5—figure supplement 2A,B*, *Figure 5—figure supplement 4A*). When comparing two samples with multiple cells in each replicate, p-values were calculated using Stratified Analysis with Nonparametric Covariable Adjustment to adjust for replicate-level variation (*Figure 4G*, *Figure 6D,E*, *Figure 5—figure supplement 1C,G*, *Figure 5—figure supplement 2D*, *Figure 5—figure supplement 3D*, *Figure 6—figure supplement 1D*, *Figure 6—figure supplement 2B,D*). When comparing two complete time curves, p-values were calculated using Welch's t-test (*Figure 3C*). When comparing datasets with three or more samples, p-values were calculated using one-way ANOVA with Sidak's multiple comparisons test (*Figure 2F,G*, *Figure 3E*, *Figure 4F,H Figure 5C,D*, *Figure 6C*, *Figure 2—figure supplement 1D*, *Figure 2—figure supplement 2D*, *Figure 2—figure supplement 3C*, *Figure 3—figure supplement 2B*, *Figure 4—figure supplement 3C,D*, *Figure 5—figure supplement 3B*, *Figure 6—figure supplement 1B*). When comparing two or more datasets over a time course or

multiple categories within a sample, p-values were calculated using two-way ANOVA with Sidak's multiple comparisons test (*Figure 1—figure supplement 1B*, *Figure 3—figure supplement 1G*, *Figure 4—figure supplement 3B*, *Figure 5—figure supplement 1B,E Figure 5—figure supplement 4C*).

## Acknowledgements

We thank F Barr, B Burke, J Martin-Serrano, and P D'Avino for antibodies, L Pelkmans and A Rai for the SRRM2-mCherry plasmid, D Ayer for the pLVX vector, and J Martin-Serrano for the HA-CHMP4C cell line. We thank B Zak for assistance in image quantification. We thank M Smith and D Wenzel for expert advice on microscopy and biochemistry, respectively. Microscopy using Leica Confocal SP8 and Nikon Automated Widefield microscopes was performed in the University of Utah Cell Imaging Core. Oligonucleotides were synthesized by the DNA/Peptide Facility, and sequencing was performed at the DNA sequencing Core Facility, all Health Sciences Center Cores at the University of Utah. Research reported in this publication also utilized the Cancer Biostatistics Shared Resource at Huntsman Cancer Institute at the University of Utah which is supported by the National Cancer Institute of the National Institutes of Health under Award Number P30CA042014. The content is solely the responsibility of the authors and does not necessarily represent the official views of NIH.

## Additional information

### Competing interests

Wesley I Sundquist: Reviewing editor, *eLife*. The other authors declare that no competing interests exist.

### Funding

| Funder | Grant reference number | Author |
| --- | --- | --- |
| National Institutes of Health | NIH R01GM112080 | Wesley I Sundquist<br>Katharine S Ullman |
| Huntsman Cancer Foundation | CRR award | Wesley I Sundquist<br>Katharine S Ullman |
| National Institutes of Health | P30CA042014 | Katharine S Ullman |

The funders had no role in study design, data collection and interpretation, or the decision to submit the work for publication.

### Author contributions

Lauren K Strohacker, Conceptualization, Data curation, Formal analysis, Supervision, Validation, Investigation, Visualization, Methodology, Writing - original draft; Douglas R Mackay, Supervision, Investigation, Methodology, Writing - review and editing; Madeline A Whitney, Formal analysis, Validation, Investigation; Genevieve C Couldwell, Formal analysis, Investigation, Writing - review and editing; Wesley I Sundquist, Katharine S Ullman, Conceptualization, Supervision, Funding acquisition, Methodology, Project administration, Writing - review and editing

### Author ORCIDs

Lauren K Strohacker (iD) https://orcid.org/0000-0001-9650-1042
Wesley I Sundquist (iD) https://orcid.org/0000-0001-9988-6021
Katharine S Ullman (iD) https://orcid.org/0000-0003-3693-2830

### Decision letter and Author response

Decision letter https://doi.org/10.7554/eLife.63743.sa1
Author response https://doi.org/10.7554/eLife.63743.sa2

## Additional files

### Supplementary files
- Supplementary file 1. Antibodies used for western blot and immunofluorescence in this study.
- Supplementary file 2. Plasmids and siRNAs used in this study.
- Supplementary file 3. Adjusted p-values calculated from this study.
- Transparent reporting form

### Data availability
All data reported in this study are included in source data files for each figure.

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
