## [Decision Letter]

**Acceptance summary:**

Abscission is the last step of cytokinesis that culminates with midbody resolution and the physical separation of the two daughter cells. The abscission checkpoint was initially described in HeLa cells as a stabilization of intercellular canals formed during late cytokinesis in response to the presence of chromatin bridges in the cleavage site, to avoid "cutting" through the DNA, which could cause genomic instability. Intercellular canal stabilization, and thus inhibition of abscission, depends on Aurora B and Aurora B-mediated phosphorylation of the ESCRT-III subunit CHMP4C. It was later found that Aurora B-dependent abscission inhibition also occurs in response to other cellular stresses, including mechanical tension at the bridge and depletion of basket nucleoporins. The present study investigates the role of cytoplasmic aggregates containing phospho-Aurora B (active), tri-phospho-CHMP4C, and the ESCRT-III component ALIX, in abscission checkpoint control. The authors implemented a robust method to enrich (up to 80%) for cells that failed to satisfy the abscission checkpoint based on previous findings from the same group, in which interference with specific NUPs prevents the recruitment of nuclear basket components (e.g. Tpr) and causes a delay in abscission. The reasons and mechanism leading to this delay remain unknown, but it has been proposed that cells actively monitor proper nuclear pore assembly before cells complete division. With this synchronization protocol at hand, the authors investigate the recruitment of midbody components under conditions of abscission delay and found that recruitment of the ESCRT components ALIX and IST1, but not CHMP4C, to early stage midbodies was delayed. Tri-phospho-CHMP4C, which binds to ALIX, was found enriched into insoluble cytoplasmic granules under conditions that prevent abscission checkpoint satisfaction. The authors named these granules "Abscission Checkpoint Bodies" or ACBs. CHMP4B and phospho-Aurora B also localize to ACBs in a "checkpoint-dependent manner". ACBs contained bona fide markers of mitotic interchromatin granules (MIGs) enriched of splicing factors that normally reside in the nucleus during interphase. Based on this, the authors propose that ACBs derive from MIGs. Interference with MIG formation caused a slight delay in abscission in HeLa cells. Other conditions that prevent abscission checkpoint satisfaction also increase the levels of ACBs, with the noticeable exception of chromatin bridges. Importantly, the increase in ACBs under these conditions is not a HeLa cell peculiarity and was also observed in non-transformed RPE1 cells, although to a reduced extent. The authors further suggest that ACB formation depends on CHMP4C. ALIX is also found at ACBs and ALIX depletion reduces the size of ACBs. Lastly, the presence of ALIX on ACBs correlated with its reduction at midbodies, suggesting that sequestration of ALIX into ACBs is a key step in the mechanism inducing a delay in abscission. Overall, this is a very interesting paper, which brings new insights into abscission control based on the very provocative concept that cytoplasmic factors regulate the timing of abscission in response to residual mitotic errors.

**Decision letter after peer review:**

Thank you for submitting your article "Abscission Checkpoint Bodies Reveal a New Facet of Abscission Checkpoint Control" for consideration by *eLife*. Your article has been reviewed by 3 peer reviewers, including Helder Maiato as the Reviewing Editor and Reviewer #1, and the evaluation has been overseen by a Reviewing Editor and Vivek Malhotra as the Senior Editor.

The reviewers have discussed the reviews with one another and the Reviewing Editor has drafted this decision to help you prepare a revised submission.

As the editors have judged that your manuscript is of interest, but as described below that additional experiments and clarifications are required before it is published, we would like to draw your attention to changes in our revision policy that we have made in response to COVID-19 (https://elifesciences.org/articles/57162). First, because many researchers have temporarily lost access to the labs, we will give authors as much time as they need to submit revised manuscripts. We are also offering, if you choose, to post the manuscript to bioRxiv (if it is not already there) along with this decision letter and a formal designation that the manuscript is "in revision at *eLife*". Please let us know if you would like to pursue this option or in case you have any questions regarding the preparation of a revised version of your manuscript.

Summary:

Abscission is the last step of cytokinesis that culminates with midbody resolution and the physical separation of the two daughter cells. The abscission checkpoint was initially described in HeLa cells as a stabilization of intercellular canals formed during late cytokinesis in response to the presence of chromatin bridges in the cleavage site, to avoid "cutting" through the DNA, which could cause genomic instability. Intercellular canal stabilization, and thus inhibition of abscission, depends on Aurora B and Aurora B-mediated phosphorylation of the ESCRT-III subunit CHMP4C. It was later found that Aurora B-dependent abscission inhibition also occurs in response to other cellular stresses, including mechanical tension at the bridge and depletion of basket nucleoporins. The present study investigates the role of cytoplasmic aggregates containing phosphor Aurora B (active), tri-phospho CHMP4C, and the ESCRT-III component ALIX, in abscission checkpoint control. The authors start by implementing a robust method to enrich (up to 80%) for cells with an "activated abscission checkpoint", based on previous findings from the same group, in which interference with specific NUPs prevents the recruitment of nuclear basket components (e.g. Tpr) and causes a delay in abscission. The reasons and mechanism leading to this delay remain unknown, but it has been proposed that cells actively monitor proper nuclear pore assembly before cells complete division. With this synchronization protocol at hand, the authors investigate the recruitment of midbody components under conditions of an "activated abscission checkpoint" (i.e. abscission delay) and found that recruitment of the ESCRT components ALIX and IST1, but not CHMP4C, to early stage midbodies was delayed. Tri-phospho CHMP4C, which binds to ALIX, was found enriched into insoluble cytoplasmic granules under conditions of "abscission checkpoint activation". The authors named these granules "Abscission Checkpoint Bodies" or ACBs. CHMP4B and phosphor Aurora B also localize to ACBs in a "checkpoint-dependent manner". ACBs contained bona fide markers of mitotic interchromatin granules (MIGs) enriched of splicing factors that normally reside in the nucleus during interphase. Based on this, the authors propose that ACBs derive from MIGs. Interference with MIG formation caused a slight delay in abscission in HeLa cells. Other conditions that cause "abscission checkpoint activation" also increase the levels of ACBs, with the noticeable exception of chromatin bridges. Importantly, the increase in ACBs after "abscission checkpoint activation" is not a HeLa cell peculiarity and was also observed in non-transformed RPE1 cells, although to a reduced extent. The authors further suggest that ACB formation depends on CHMP4C, although this might be indirect. ALIX is also found at ACBs and ALIX depletion reduces the size of ACBs. Lastly, the presence of ALIX on ACBs correlated with its reduction at midbodies, suggesting that sequestration of ALIX into ACBs is a key step in the mechanism inducing a delay in abscission.

Overall, this is a very interesting paper, which brings new insights into abscission control based on the very provocative idea that cytoplasmic factors regulate the timing of abscission in response to mitotic errors. However, it was consensual amongst the 3 reviewers that compelling evidence is still lacking to justify the claim that ACBs are formed in response to conditions that delay abscission, or that they regulate abscission timing. In particular, there are gaps in the interpretation of the results that should be addressed experimentally and additional data must be provided to support the main conclusions, as outlined below.

Essential revisions:

1. One of the main concerns relates with the interpretation of the concept behind the Abscission Checkpoint. While evidence from several works exist in support of conditions that cause a delay in abscission, and thus support the existence of checkpoint control, the idea that these conditions "activate" or "induce" this abscission checkpoint is misleading. As the authors properly cite, the concept of checkpoints proposed by Hartwell and Weinert in 1989 implies the existence of surveillance mechanisms that are active by default and are external to the process being monitored. Only in such way errors can be effectively monitored and cell cycle progression (including completion of abscission) can be delayed to allow for eventual correction. Thus, checkpoints are satisfied, not activated. In the present work, the authors suggest that they can synchronously "activate" the abscission checkpoint, which could alternatively be interpreted as they can synchronously cause a delay in abscission that prevents its satisfaction due to unmet conditions supervised by the abscission checkpoint. In this case, the default condition used in this work was the constitutive depletion of some NUPs, whose link to mitotic errors remains unknown, i.e. why and how does this cause a delay in completing abscission? This becomes even more complicated if other prototype mitotic errors causing an abscission checkpoint response, such as lagging chromatin bridges, do not seem to cause an increase in ACBs, suggesting that ACB formation might be an epiphenomenon, rather than a key component of the mechanism underlying the abscission checkpoint. As so, the authors must clarify the differences between the cellular response to NUPs depletion and lagging chromatin. One possibility is that the conditions underlying NUPs depletion prevent proper nuclear pore function and could for example be a mere consequence of partially defective nuclear import, since nuclear speckle components are normally re-imported after nuclear envelope reformation. The authors try to rule out this possibility with co-depletion of CHMP4C, which reverses the formation of ACBs. They should nevertheless show by western blot that this triple depletion allows sufficient depletion of NUPs to the level required to prevent checkpoint satisfaction. Another possibility would be to show that depleting other NUPs that do not affect checkpoint response does not trigger the formation of ACBs. Lastly, inhibiting Aurora B activity should dissolve the ACBs, since it would impair CHMP4C phosphorylation. Is it the case?

2. Related to the previous point, the aphidicolin experiments are important to support the conclusion that ACBs are caused by conditions that prevent checkpoint satisfaction, and not by defective nuclear import. However, it is possible that aphidicolin does not delay abscission under the conditions used. Instead, aphidicolin could just delay S-phase and thus all subsequent cell cycle stages without affecting the duration of abscission itself. To test this, the fraction of midbody-stage cells with and without ACBs should be determined for the entire duration of the time course after aphidicolin addition, and not only for the interval between 12 and 18 hours after thymidine release (Figure 5 supplement 1B-D). This would also reveal if ACBs appear in unchallenged cells earlier in the cell cycle, which would be useful to understand their origin.

3. On the true existence of a "checkpoint" in the sense of a truly external mechanism that oversees abscission vs. the lack of a key structural component necessary to complete abscission. Given the observed accumulation of ACBs and their proposed origin from MIGs, it could be that some splicing event of a rate limiting mRNA/protein necessary to complete abscission is flawed. Given that the authors can synchronize cells in such delayed abscission condition, it would be instrumental to investigate whether splicing (by RT-PCR) and production of key proteins (by western blot) involved in abscission is taking place normally.

4. Are ACBs functionally involved in the abscission checkpoint? ACBs appear upon conditions that prevent checkpoint satisfaction, disappears when the checkpoint component CHMP4C is depleted and correlates with decreased levels of ALIX in the intercellular bridge. These are interesting but correlative evidence for a function of ACBs in the checkpoint. The authors conclude that ACBs regulate abscission timing (Figure 4), since stimulating their formation/maintenance (through DYRK3 or CLK1/2 inhibition) delays abscission in normal cells. This raises several questions. First, the abscission delay is rather modest and could only partially explain the abscission delay observed upon conditions that preclude checkpoint satisfaction. Second, this is different from previous studies (e.g. PMID 27126587), which, on the contrary, it was found that CLK inhibition accelerated abscission. The authors argue that they used a more specific inhibitor and for a shorter period of time. A simple test would be to use the same treatment as in PMID 27126587 (5h, pan-CLK inhibitor) and to show that in these conditions ACBs are dissolved. Third and more importantly, one would be fully convinced that ACBs regulate the abscission checkpoint if abscission timing is restored to normal upon conditions that prevent checkpoint satisfaction when ACBs are experimentally dissolved. For instance, overexpression of CLK (but not a kinase dead CLK) was previously found to dissolve MIGs (PMID 11827980). Would the abscission delay be abolished by ACBs dissolution in NUP-depleted cells? What is the effect of CLK inhibition in RPE1 cells? These or equivalent experiments would greatly enhance the mechanistic implications and of ACBs and their general role in the abscission checkpoint.

5. The existence of ACBs (positive for pppCHMP4C, pAuroraB and CHMP4B) is demonstrated in fixed cells that have been pre-permeabilized before fixation. Nobody has reported so far the existence of CHMP4B aggregates in the cytoplasm upon conditions that prevent checkpoint satisfaction. In addition, the midbody staining of CHMP4B is unusual (Figure 3D), perhaps as a consequence of pre-permeabilization. Can the authors find a way to show that ACBs exist in NUP-depleted cells without pre-permeabilization (which might induce phase separation in these cells)? Showing ACBs in live cells (no fixation) using CHMP4B-LAP-GFP, CHMP4C-LAP-GFP and speckle markers (e.g. SC35-YFP as in PMID 20926517 Figure 2) would be even better than after fixation. One would expect cytoplasmic dots in live cells upon NUP depletion using these makers.

---

## [Author Response]

Essential revisions:1. One of the main concerns relates with the interpretation of the concept behind the Abscission Checkpoint. While evidence from several works exist in support of conditions that cause a delay in abscission, and thus support the existence of checkpoint control, the idea that these conditions "activate" or "induce" this abscission checkpoint is misleading. As the authors properly cite, the concept of checkpoints proposed by Hartwell and Weinert in 1989 implies the existence of surveillance mechanisms that are active by default and are external to the process being monitored. Only in such way errors can be effectively monitored and cell cycle progression (including completion of abscission) can be delayed to allow for eventual correction. Thus, checkpoints are satisfied, not activated. In the present work, the authors suggest that they can synchronously "activate" the abscission checkpoint, which could alternatively be interpreted as they can synchronously cause a delay in abscission that prevents its satisfaction due to unmet conditions supervised by the abscission checkpoint.

We agree and thank the reviewers for making this point. We have clarified the language throughout to make this important distinction.

In this case, the default condition used in this work was the constitutive depletion of some NUPs, whose link to mitotic errors remains unknown, i.e. why and how does this cause a delay in completing abscission? This becomes even more complicated if other prototype mitotic errors causing an abscission checkpoint response, such as lagging chromatin bridges, do not seem to cause an increase in ACBs, suggesting that ACB formation might be an epiphenomenon, rather than a key component of the mechanism underlying the abscission checkpoint. As so, the authors must clarify the differences between the cellular response to NUPs depletion and lagging chromatin. One possibility is that the conditions underlying NUPs depletion prevent proper nuclear pore function and could for example be a mere consequence of partially defective nuclear import, since nuclear speckle components are normally re-imported after nuclear envelope reformation. The authors try to rule out this possibility with co-depletion of CHMP4C, which reverses the formation of ACBs. They should nevertheless show by western blot that this triple depletion allows sufficient depletion of NUPs to the level required to prevent checkpoint satisfaction. Another possibility would be to show that depleting other NUPs that do not affect checkpoint response does not trigger the formation of ACBs. Lastly, inhibiting Aurora B activity should dissolve the ACBs, since it would impair CHMP4C phosphorylation. Is it the case?

a) As suggested by the reviewers, we have performed western blots to demonstrate that both Nup153 and Nup50 are efficiently knocked down in the triple depletion (as is CHMP4C). These data are now presented in the new Figure 5—figure supplement 5, and they clearly show that Nup153 and Nup50 are knocked down to levels that induce abscission delays in the presence of CHMP4C. Importantly, under these conditions, CHMP4C depletion significantly decreases ACB-formation and prevents midbody arrest (Figure 5D). We also refer reviewers to a previous publication, PMID: 30181294, in which we showed that midbody arrest is rescued by re-expression of exogenous CHMP4C following depletion of endogenous CHMP4C.

b) As suggested, we also performed experiments with AurB inhibitors (AurBi), and have enclosed the data for the reviewers’ information (see Author response image 1). We found that Aurora B inhibition does not completely dissolve ACBs (1A) but does decrease ACB numbers (1B). An important point is that although the AurB inhibitor allosterically prevents AurB from phosphorylating its targets, it does not reverse AurB phospho-status. When we treat with AurBi, we find that phospho-AurB signal at the midbody decreases >10-fold whereas the phospho-AurB signal in ACBs remains unchanged (1C). It seems likely that phosphatases at the midbody dephosphorylate AurB to destabilize the midbody and eventually enable abscission, whereas AurB remains phosphorylated in ACBs. We hypothesize that disassembly of ACBs requires an active process that is likely to involve recruitment of additional factors. Thus, this line of experiments generated useful information but we have not included it in the revised manuscript because it did not address the question that prompted it.

**Author response image 1. sa2fig1:** AurB inhibition decreases but does not completely dissolve ACBs. (A) Quantification of midbody-stage cells (Intact) and recently-abscised midbody pairs (Cut) with ACBs marked by α-pAurB after treatment with siCon or siNups and thymidine synchronization. With and without 1 h treatment with AurBi added at 15 h, 1 h prior to harvest at 16 h. N=500 pairs per condition from n=5 biological replicates. (B) Number of ACBs per midbody pair (marked by α-pAurB) after treatment as in A (siNups only) n=3 biological replicates. (C) pAurB intensity per area in ACBs or at the midbody after treatment as in A (siNups only) n=4 biological replicates.

Overall, the strong requirement for CHMP4C and our finding that two other prototypic errors surveilled by the abscission checkpoint (replication stress and increased intercellular tension) result in ACB formation supports the conclusion that ACBs are an abscission checkpoint mechanism. We note that lagging chromatin, which does not support ACB formation, creates a unique cellular structure that is located proximal to the abscission site. Thus, we hypothesize that unlike the other known signals that maintain NoCut activation, lagging chromatin maintains NoCut signals without promoting ACB formation, although these different upstream scenarios share a reliance on Aurora B and ultimately converge on events at the midbody that delay abscission.

2. Related to the previous point, the aphidicolin experiments are important to support the conclusion that ACBs are caused by conditions that prevent checkpoint satisfaction, and not by defective nuclear import. However, it is possible that aphidicolin does not delay abscission under the conditions used. Instead, aphidicolin could just delay S-phase and thus all subsequent cell cycle stages without affecting the duration of abscission itself. To test this, the fraction of midbody-stage cells with and without ACBs should be determined for the entire duration of the time course after aphidicolin addition, and not only for the interval between 12 and 18 hours after thymidine release (Figure 5 supplement 1B-D). This would also reveal if ACBs appear in unchallenged cells earlier in the cell cycle, which would be useful to understand their origin.

To address this concern, we performed live-imaging experiments in which we measured the time from midbody formation to scission (i.e., abscission time) in DMSO vs aphidicolin-treated cells. These data are presented in the new Figure 5—figure supplement 1C. Importantly, we observe that aphidicolin treatment delays specifically abscission timing by ~21 minutes, consistent with a previously published result (PMID: 25904336). Thus, the duration of abscission is significantly delayed by aphidicolin treatment.

3. On the true existence of a "checkpoint" in the sense of a truly external mechanism that oversees abscission vs. the lack of a key structural component necessary to complete abscission. Given the observed accumulation of ACBs and their proposed origin from MIGs, it could be that some splicing event of a rate limiting mRNA/protein necessary to complete abscission is flawed. Given that the authors can synchronize cells in such delayed abscission condition, it would be instrumental to investigate whether splicing (by RT-PCR) and production of key proteins (by western blot) involved in abscission is taking place normally.

We agree with the reviewers that an RNA splicing event could indeed be required for abscission to take place, and we view this as one of the possible mechanisms by which checkpoint signaling could delay abscission (and this possibility is now discussed explicitly). Whether or not this intriguing possibility is correct, however, we believe that assaying RNA splicing and identifying a (putative) regulated splicing target is beyond the scope of this paper.

4. Are ACBs functionally involved in the abscission checkpoint? ACBs appear upon conditions that prevent checkpoint satisfaction, disappears when the checkpoint component CHMP4C is depleted and correlates with decreased levels of ALIX in the intercellular bridge. These are interesting but correlative evidence for a function of ACBs in the checkpoint. The authors conclude that ACBs regulate abscission timing (Figure 4), since stimulating their formation/maintenance (through DYRK3 or CLK1/2 inhibition) delays abscission in normal cells. This raises several questions. First, the abscission delay is rather modest and could only partially explain the abscission delay observed upon conditions that preclude checkpoint satisfaction. Second, this is different from previous studies (e.g. PMID 27126587), which, on the contrary, it was found that CLK inhibition accelerated abscission. The authors argue that they used a more specific inhibitor and for a shorter period of time. A simple test would be to use the same treatment as in PMID 27126587 (5h, pan-CLK inhibitor) and to show that in these conditions ACBs are dissolved. Third and more importantly, one would be fully convinced that ACBs regulate the abscission checkpoint if abscission timing is restored to normal upon conditions that prevent checkpoint satisfaction when ACBs are experimentally dissolved. For instance, overexpression of CLK (but not a kinase dead CLK) was previously found to dissolve MIGs (PMID 11827980). Would the abscission delay be abolished by ACBs dissolution in NUP-depleted cells? What is the effect of CLK inhibition in RPE1 cells? These or equivalent experiments would greatly enhance the mechanistic implications and of ACBs and their general role in the abscission checkpoint.

a) We agree that the abscission delay under DYRK3 or CLK1/2 inhibition is modest. We also observe that the enhancement of ACB formation under these conditions is modest compared to what we see with Nup depletion (likely because this is just one arm of a complex signaling pathway). Multiple mechanisms are known to halt abscission when the checkpoint is active, so even under the best of circumstances, inducing just one of these mechanisms would not be expected to recreate the strength of a coordinated checkpoint response. Therefore, in our view, the effect on abscission seen with DYRK3 or CLK1/2 inhibition – in the absence of other signals and components- provides significant support for our model.

b) As suggested, we compared our specific CLK1/2 inhibitor to the previously-used pan-CLK inhibitor (see Author response image 2). In parallel experiments, we find that the specific CLK1/2 inhibitor induces mild abscission arrest and ACB formation while the pan-CLK inhibitor has no effect either on abscission timing or ACB formation. We conclude that the pan-CLK inhibitor may have a different potency or that the lack of effects could potentially reflect opposing roles of CLK family members. Regardless, the experiments did not change our conclusions about CLK1.

**Author response image 2. sa2fig2:** Specific CLK1/2 inhibition delays abscission timing. (A) Quantification of midbody-stage cells after 5 h asynchronous treatment with DMSO, 0.5 – 10 µM Pan-CLK inhibitor (Pan-CLKi, TG003, Sigma), or 1 µM specific CLK1/2 inhibitor (CLK1i, 534350, Millipore). N=3200 cells from n=5 biological replicates. (B) Quantification of midbody-stage cells with foci marked by SC35 antibody after treatment as in A. N=800 midbody-stage cells from n=4 biological replicates. (C) Quantification of midbody-stage cells with ACBs (both larger and more numerous than MIGs, and present in later midbody-stage cells) marked by SC35 antibody after treatment as in A. N=800 midbody-stage cells from n=4 biological replicates.

c) Following the reviewers’ suggestion, we overexpressed CLK1 (PMID 11827980, PMID 8617202) in a “checkpoint unsatisfied” background and tested the effect on ACB frequency and numbers and midbody frequencies. We find that overexpressing WT CLK1 consistently and significantly decreases midbody-stage cells and correspondingly decreases ACB numbers. ACB disassembly corresponds to increased nuclear signal, further supporting the conclusion that CLK1 encourages ACB factors to leave this site and become available for nuclear entry. These data are now presented in new Figures 4H-I and Figure 4—figure supplement 4.

In summary, we find that CLK1 inhibition increases ACB numbers and frequency, and correspondingly increases midbody numbers. Conversely, overexpressing CLK1 decreases ACB numbers, and correspondingly decreases midbody-stage cells. While these effects are modest, they are proportional to the extent that we can manipulate ACB formation/dissolution and the effects of manipulating this regulatory mechanism in isolation, and collectively support the conclusion that ACBs contribute directly to abscission regulation.

5. The existence of ACBs (positive for pppCHMP4C, pAuroraB and CHMP4B) is demonstrated in fixed cells that have been pre-permeabilized before fixation. Nobody has reported so far the existence of CHMP4B aggregates in the cytoplasm upon conditions that prevent checkpoint satisfaction. In addition, the midbody staining of CHMP4B is unusual (Figure 3D), perhaps as a consequence of pre-permeabilization. Can the authors find a way to show that ACBs exist in NUP-depleted cells without pre-permeabilization (which might induce phase separation in these cells)? Showing ACBs in live cells (no fixation) using CHMP4B-LAP-GFP, CHMP4C-LAP-GFP and speckle markers (e.g. SC35-YFP as in PMID 20926517 Figure 2) would be even better than after fixation. One would expect cytoplasmic dots in live cells upon NUP depletion using these makers.

We thank the reviewers for giving us an opportunity to clarify this point. We did, in fact, initially show that each of the above markers localizes to ACBs in non-permeabilized cells (pppCHMP4C: Figure 3A; SC35: Figure 4B; pAurB: Figure 4—figure supplement 1A, CHMP4B: Figure 4—figure supplement 1B) before using pre-permeabilization to aid in staining clarity for ease of quantification. We have clarified this point in the text. We also agree that pre-permeabilization does disrupt midbody staining for some proteins (e.g., CHMP4B) more than others (e.g., pAurB). Finally, we have included new data in Figure 4C showing that in live cells, SRRM2-mCherry localizes strongly to ACBs under checkpoint active conditions.